# Generation, localization and functions of macrophages during the development of testis

Emmi Lokka[1,2,3], Laura Lintukorpi[1,2], Sheyla Cisneros-Montalvo [1], Juho-Antti Mäkelä [1], Sofia Tyystjärvi [1,2], Venla Ojasalo[1,2], Heidi Gerke[1,2], Jorma Toppari[1,4], Pia Rantakari [1,2,5✉] & Marko Salmi [1,3,5✉]

In the testis, interstitial macrophages are thought to be derived from the yolk sac during fetal development, and later replaced by bone marrow-derived macrophages. By contrast, the peritubular macrophages have been reported to emerge first in the postnatal testis and solely represent descendants of bone marrow-derived monocytes. Here, we define new monocyte and macrophage types in the fetal and postnatal testis using high-dimensional single-cell analyses. Our results show that interstitial macrophages have a dominant contribution from fetal liver-derived precursors, while peritubular macrophages are generated already at birth from embryonic precursors. We find that bone marrow-derived monocytes do not substantially contribute to the replenishment of the testicular macrophage pool even after systemic macrophage depletion. The presence of macrophages prenatally, but not postnatally, is necessary for normal spermatogenesis. Our multifaceted data thus challenge the current paradigms in testicular macrophage biology by delineating their differentiation, homeostasis and functions.

---

[1] Institute of Biomedicine, University of Turku, Turku FI-20520, Finland. [2] Turku Bioscience Centre, University of Turku and Åbo Akademi University, Turku FIN-20520, Finland. [3] MediCity Research Laboratory, University of Turku, Turku FI-20520, Finland. [4] Department of Pediatrics, Turku University Hospital, Turku FI-20520, Finland. [5] These authors contributed equally: Pia Rantakari, Marko Salmi. ✉email: pia.rantakari@utu.fi; marko.salmi@utu.fi

Tissue-resident macrophages are a heterogeneous population of immune cells involved in the tissue homeostasis, remodeling, and immune surveillance[1–5]. Recent studies have revealed diverse developmental waves, which generate embryonic and adult macrophages in different tissues[6–9]. The mechanisms that regulate the differentiation of tissue-resident macrophages in the local tissue microenvironment are currently a topic of intensive research[10–16].

In the testis tissue-resident macrophages make up a large proportion of the somatic cell population and are the main leukocyte type[17,18]. Testicular tissue-resident macrophages have been proposed to perform several functions during development and adulthood[19–22]. During the embryonic period, they regulate vascularization and morphogenesis[23]. Early studies with postnatal mice, mainly analyzing isolated cell types in culture, implied the importance of testicular macrophages to steroidogenesis[19,24–27]. Under homeostatic conditions in vivo tissue-resident macrophages were reported to provide a suitable tissue niche that supports Leydig cell function[24–26,28], and assist the differentiation of spermatogonia[29]. Testicular macrophages have also been implied in the maintenance of the immunosuppressive environment in the testis[20,28,30].

Two tissue-resident macrophage populations have been identified in the testis. Peritubular macrophages lie on the surface of seminiferous tubules next to the peritubular myoid cells, and interstitial macrophages are located within the testicular interstitium in close contact with Leydig cells[18,23,29,31]. Currently, it is thought that the peritubular macrophages are derived from circulating, bone-marrow-derived monocytes[31], and that the interstitial macrophages originate from fetal, yolk-sac-derived cells but are then replaced by bone-marrow-derived macrophages after birth[23]. However, no comprehensive high-resolution single-cell analyses on the development of testicular macrophage landscape have been reported.

Here, we examine the developmental origin and turnover of tissue-resident macrophages in the testis using in-depth high-dimensional single-cell proteomic analysis combined with studies using cell-fate mapping, macrophage depletion, and macrophage deficient mice. We demonstrate that under normal physiological conditions, tissue-resident macrophages in the adult testis are derived from embryonic precursors and that postnatally the macrophages are dispensable for spermatogenesis. These findings profoundly alter our understanding of the heterogeneity, origins, and functions of macrophages in the testis, and reveal their unique adaptations when compared to tissue-resident macrophages at other locations.

## Results

**Fetal macrophages infiltrate the testes anlage**. To analyze the overall leukocyte landscape in the fetal testis by flow cytometry, we first gated CD45+ cells and then analyzed the presence of macrophage (F4/80+) populations (for the pregating see Supplementary Fig. 1a). In most fetal organs, F4/80Hi cells correspond to the yolk-sac-derived and F4/80Int cells to the fetal liver-derived macrophages[14,32–35]. We found that at embryonic day (E) 14.5, the F4/80Hi macrophages outnumbered the F4/80Int cells in the testis, whereas at E16.5 and in newborns the F4/80Int cells had become the dominant population (Fig. 1a, Supplementary Fig. 2a). Of note, F4/80Hi cells also expressed lower levels of CD45 compared to F4/80Int cells. At E16.5 the F4/80Hi cells were mostly CD115+, CD206+, and Ly6CLow, whereas the F4/80Int cells expressed only low levels of CD115, and were CD206− and Ly6CHi (Fig. 1b; Supplementary Fig. 2b). The reciprocal expression of CD206 and Ly6C persisted on the two macrophage populations till birth (Fig. 1c; Supplementary Fig. 2c). F4/80 can

also be expressed by dendritic cells and eosinophils[36], but in F4/80Hi or F4/80Int populations of E16.5 or newborn animals we found only small numbers of CD11cHi dendritic cells and modest numbers of SiglecFHi eosinophils (more substantial among F4/80Int cells at birth; Supplementary Fig. 2d, e).

To interrogate the putative yolk sac origin of F4/80Hi cells in the testis, we depleted all yolk sac macrophages with a single pulse injection of CD115 antibody (against CSF1R, clone AFS98) into pregnant dams at E6.5 using a previously validated protocol[9,37–39] (Fig. 1d; Supplementary Fig. 2f). By analyzing the cell frequencies and absolute cell numbers in E17.5 testes, we found that the CD115 antibody depleted very efficiently F4/80Hi cells without having any effect on the F4/80Int cells (Fig. 1d; Supplementary Fig. 2g). These analyses thus show that the yolk-sac-derived macrophages are F4/80Hi also in the testis.

High-dimensional mass cytometric analyses of testis macrophages first became feasible at birth due to organ size. Unsupervised clustering of viable CD45+CD11b+F4/80+ cells with viSNE algorithm revealed the presence of five distinct myeloid cell populations at birth (Fig. 1e, f; Supplementary Fig. 2h, for the gating strategy see Supplementary Fig. 1f). In addition to substantial SiglecFHi (putative eosinophils) and small CD11c+MHC IIHi (putative dendritic cells) clusters, three cell clusters positive for core macrophage markers CD64 and CX3CR1 were identified. The largest of them was Ly6C+, and contained both Ly6CLow and Ly6CHi subpopulations (Fig. 1e, f). A second CD64+CX3CR1+F4/80+ cluster was brightly positive for CD206, whereas the third one was characterized by CD206 negativity (Fig. 1e, f Supplementary Fig. 2h). Notably, all testicular monocyte/macrophage cell populations were completely MHC II− at birth.

In in silico lineage tracing, in which the relatedness of cells is inferred from their similarity to the nearest neighbors[40], we found that the two identified monocyte populations (Ly6CHi and Ly6CLow cells) showed high connectivity to each other in newborns. The Ly6CLow cells also had trajectories to the CD206− macrophages, which in turn were closely associated with CD206+ cells (Fig. 1g; Supplementary Fig. 2i). Collectively these data suggest that testicular leukocytes expressing the highest level of Ly6C likely represent recent monocyte immigrants. The Ly6CLow cells, also expressing CD64, are apparently monocytes more differentiated to the direction of F4/80HiCD64Hi macrophages, which are first CD206−, but can subsequently upregulate CD206. In addition to monocyte-derived CD206+ macrophages, monocyte-independent F4/80Hi yolk-sac-derived macrophages also likely directly contribute to the CD206+ macrophage population, especially during the early fetal development.

**Macrophage diversity increases in the postnatal testes**. To avoid exclusion of any myeloid cell population in analyses of the postnatal testis from 2-, 5-, and 14-week-old mice, we included all CD45+CD11b+ cells in our kinetic high-dimensional mass cytometric studies (for the gating strategy see Supplementary Fig. 1g). We immediately observed that F4/80 and CD64, used for specifying the subpopulations in the previous reports[23,31], did not allow a clear-cut separation of the macrophage types (Fig. 2a). Notably, a large MHC II+ population gradually evolved in the testis after birth, and in fact, MHC II and CD206 expression allowed clear delineation of CD206+MHC II−, CD206−MHC II+, and CD206−MHC II− macrophage subpopulations postnatally (Fig. 2b).

Unsupervised FlowSOM analyses[41] further divided the CD206+MHC II− macrophage type into CD115+ and CD115− subpopulations (Fig. 2c), which also showed differences in the expression of TIM3 and CD43 (Supplementary Fig. 3a–c). On the

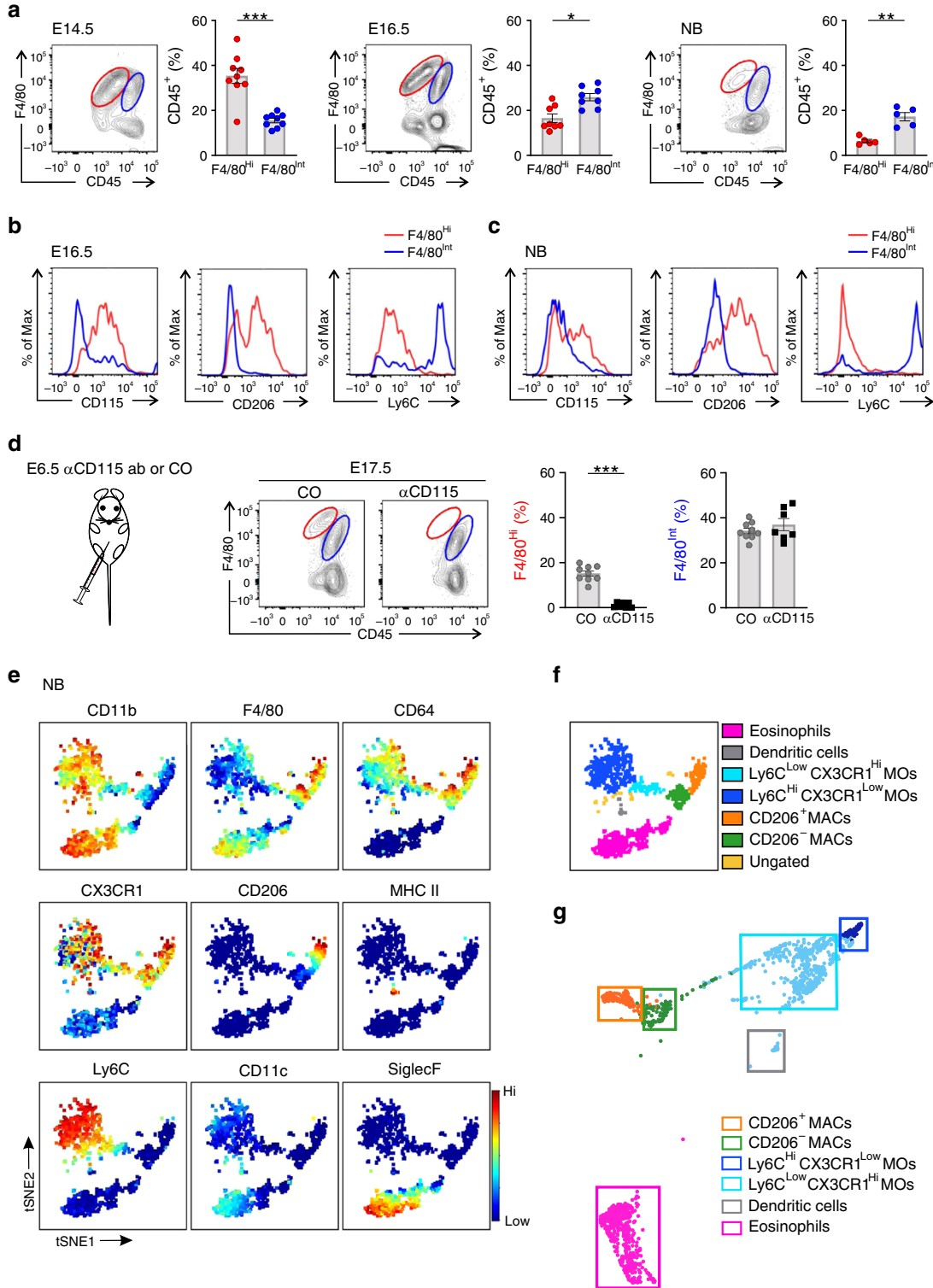

other hand, CD206−MHC II+ and CD206−MHC II− macrophages each comprised only one population also in the FlowSOM analyses (Fig. 2c). These cells expressed intermediate levels of F4/80, CD64, and CX3CR1, but very little or no CD115, TIM3, TIM4, or Siglec1 (Fig. 2a, b; Supplementary Fig. 3a–c). Moreover, unsupervised analyses also identified a very small CD206+MHC II+ macrophage population in the testis (Fig. 2a–c).

The frequencies of the four main macrophage populations showed characteristic changes during the postnatal testicular maturation (Fig. 2d). In 2-week-old mice the CD206+MHC II−

and CD206−MHC II− populations dominated. In 14-week-old mice, in contrast, the frequency of these two populations was strongly reduced, while the CD206−MHC II+ population represented about 60% of all myeloid cells. The rare CD206+MHC II+ population did not contribute more than 5% to myeloid cells at any timepoint analyzed.

Distinct Ly6C^Hi and Ly6C^Low cell populations were present in 2–14-week-old testes (Fig. 2b). The Ly6C^Hi cells were negative for MHC II and thus likely represent recent monocyte immigrants. Of the Ly6C^Low populations, one co-expressed Ly6G and CD43,

**Fig. 1 Yolk-sac-derived and fetal liver-derived macrophage populations are present in the embryonic testis. a** Flow cytometric (FACS) analyses of testicular macrophage populations in embryonic (E) and newborn (NB) wild-type (WT) mice. Shown are representative FACS plots and frequencies of F4/80[Hi] (red) and F4/80[Int] (blue) testicular macrophage populations at the indicated timepoints. **b, c** Representative histograms of CD115, CD206, and Ly6C expression in F4/80[Hi] and F4/80[Int] testicular macrophage populations in **b** E16.5 and **c** NB WT mice. **d** Analysis of F4/80[Hi] and F4/80[Int] testicular macrophages in E17.5 WT embryo treated with blocking CD115 antibody or control IgG at E6.5. Experimental outline, representative FACS plots, and frequencies of F4/80[Hi] and F4/80[Int] are shown. **e** t-SNE maps from mass cytometric analyses displaying the expression of the indicated antigens in randomly sampled live, single CD45[+]CD11b[+]F4/80[+] cells in testes of NB WT mice. The scale bar indicates the expression level of a given antigen from low (blue) to high (red). **f** Manual clustering of cell populations based on the expression of known myeloid markers shown in (**e**). MAC macrophage, MO monocyte. **g** Unsupervised hierarchical X-shift clustering (nearest neighbor) illustration of macrophage populations from (**e**). The colored boxes show the location of manually gated cell populations based on the expression analyses of myeloid cell-selective markers (shown in Supplementary Fig. 2i). In the quantifications (**a, d**), each dot represents a pool of 2–8 testes from 1–4 mice (E14.5, $n = 9$, and E16.5, $n = 8$ pools), or both testes of one mouse (E17.5, CO; $n = 9$, CD115; $n = 7$ and NB, $n = 5$ mice). Data are presented as mean ± SEM (*$p < 0.05$, **$p < 0.01$, ***$p < 0.001$, Two-tailed Mann–Whitney U test). All data are from 2–3 independent experiments. Mass cytometry data (**e–g**) is from 6–10 testes each pooled from 3–5 mice from two independent experiments. Source data are provided as a Source Data file.

and thus likely represents granulocytes, while the other was Ly6G[−]CD43[Hi], and likely represents tissue-infiltrating monocytes differentiating towards macrophages or dendritic cells (Supplementary Fig. 3a–c). The frequency of Ly6C[Hi] and Ly6C[Low] monocytes in the testis remained very low in 2-, 5-, and 14-week-old mice (Fig. 2e). In addition, a small SiglecF[+] cell cluster (presumably eosinophils) and small CD11c[+] clusters (tentative dendritic cells) were identifiable among the myeloid cells in the postnatal testis (Supplementary Fig. 3a–c).

Taken together, the unsupervised high-dimensional analyses reveal the presence of main CD206[+]MHC II[−] (with two distinct subpopulations) and CD206[−]MHC II[−] macrophage populations, which are declining during aging, and an emerging CD206[−]MHC II[+] population in the testis of prepubertal, juvenile and adult mice.

**Associations of different testicular macrophage types.** In silico lineage tracing of 2–14-week mice, testicular Ly6C[+] monocytes showed clear connection neither to CD206[−]MHC II[+] macrophages, which are suggested to be derived from bone marrow monocytes[31], nor to the other macrophage populations at any timepoint analyzed (Fig. 2f; Supplementary Fig. 4a–c). On the other hand, the CD206[−]MHC II[−] macrophages still had clear connectivity to CD206[+]MHC II[−] cells in 2-week-old mice. At the 5-week timepoint, CD206[−]MHC II[−]cells, in contrast, had the strongest associations with CD206[−]MHC II[+] macrophages, and by the 14-week timepoint, the double negative macrophages had practically disappeared from the testis (Fig. 2f). Interestingly, in 2-week-old mice the double negative macrophage population (and subpopulations of CD206[+]MHC II[−] cells) had the highest frequency of proliferating Ki67[+] cells, while different Ly6C[+] monocytic clusters remained virtually non-proliferative (Supplementary Fig. 3c). The proliferation of testicular macrophages was substantially decreased in 5-week-old mice. Pooled in silico lineage tracing data from the fetal and postnatal timepoints are compatible with a scenario that CD206[−]MHC II[−] cells infiltrate the testes during the fetal period (presumably as fetal-derived Ly6C[+] monocytes) and first give rise to CD206[+]MHC II[−] cells in young mice and subsequently differentiate to CD206[−]MHC II[+] macrophages during later postnatal life. The role of postnatal bone-marrow-derived monocytes in the generation of testicular macrophages, in contrast, appears to be minimal.

**Distinct functions and localization of testis macrophages.** To allow dimensionality reduction we took advantage of the discriminative power of CD206 and MHC II in defining the main tissue-resident macrophage populations in the testis. The main macrophage populations (CD206[+]MHC II[−], CD206[−]MHC II[−],

and CD206[−]MHC II[+]) were identifiable in the testes by fluorimetric assays, and they were free of any significant dendritic cell (CD11c[Hi]) or eosinophil (SiglecF[+]) contamination (Fig. 3a; Supplementary Fig. 5a, b; for the gating strategy see Supplementary Fig. 1b and 1h).

We then tested the scavenging function of the testicular macrophage populations. When isolated testicular myeloid (CD11b[+]) cells were allowed to engulf high molecular weight fluorescent dextran under ex vivo conditions, CD206[+]MHCII[−] cells were the most active ones (Fig. 3b). However, the other macrophage subtypes were also capable of specific endocytosis supporting their genuine macrophage nature. To compare the functions of these three macrophage types in vivo, we injected intravenously fluorescent dextran, immunocomplexes, and acetylated LDL into 5-week-old mice. After a 1–2-h circulation time, quantitative FACS analyses of the whole testis, CD206[+]MHC II[−] macrophages showed clearly a superior capacity to bind/ingest these cargoes when compared to the other two macrophage types (Fig. 3c–e). In situ analyses of the testis verified that CD206[+] cells had bound dextran, whereas MHC II[+] cells were largely devoid of it (Fig. 3f, g).

Macrophages are known to lodge in two distinct compartments in the testis[29,31]. The peritubular macrophages with elongated morphology embrace the gamete-producing seminiferous tubules, whereas the interstitial macrophages reside in the testicular interstitium between the tubules. When using CD206 and MHC II as the optimized markers for the testicular macrophage populations in intact wild-type mice, we found a clear association of CD206[+]MHC II[−] cells with the CD31[+] vasculature in the interstitial tissue (Fig. 3h). The CD206[−]MHC II[+] cells, in contrast, were less frequently associated with the vasculature but were typically in close contact with the tubules (Fig. 3h). In quantitative analyses, 82.4% of the vessel-associated (direct contact) macrophages were CD206[+] cells and 17.6% were MHCII[+] cells (altogether 142 macrophages from three mice were scored). In this context, it is important to note that peritubular F4/80[+] macrophages were in fact present in tubule periphery, although at low numbers, in 1- and 2-week-old mice and likely already in newborns (Supplementary Fig. 5c, d). Collectively these data indicate that the peritubular macrophages are correctly localized in the testis well before the previously assumed 2-week timepoint[31], although they do not yet express MHC II at these early timepoints. Moreover, the preferentially perivascular localization of CD206[+]MHC II[−] macrophages may be associated with their superior scavenging capacity for blood-borne cargoes.

**Fetal-derived macrophages persist until adulthood in testes.** To track the origin of testicular macrophages, we used cell-fate

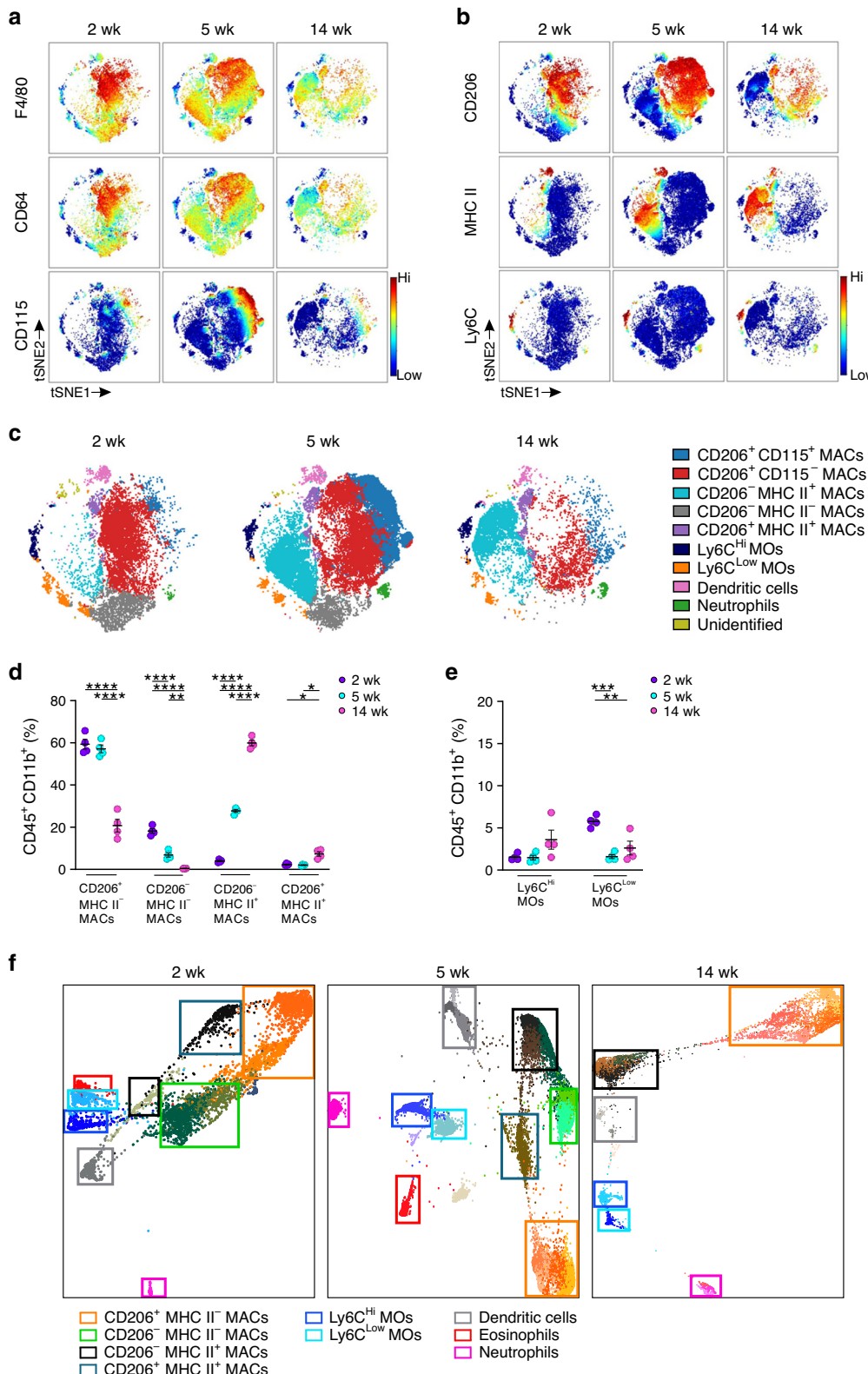

mapping and depletion experiments and mice lacking defined monocyte and macrophage subsets. We first traced yolk-sac-derived macrophages (and their progeny) using the canonical $CSF1R^{Mer-iCre-Mer};R26R-EYFP$ reporter mice[42]. When the CSF1R-reporter mice were treated by a single tamoxifen injection at E8.5, the progeny cells were found within F4/80$^{Hi}$ macrophages in the testis of E16.5 mice (Fig. 4a). When the

yolk-sac-derived macrophages were fate-mapped in 5-week-old CSF1R-reporter mice, the YFP$^+$ cells were found in all three macrophage subpopulations, albeit their frequency was highest in CD206$^+$MHC II$^-$ cells (Fig. 4b). We also used another genetic lineage tracing model, which selectively labels yolk-sac-derived macrophages[14,32,43]. Tamoxifen pulsing of $CX3CR1^{CreERT2};R26R-EYFP$ mice at E13.5 led to preferential

**Fig. 2 Kinetically evolving macrophage and monocyte populations in the testis after birth. a, b** t-SNE maps displaying the expression of randomly sampled testicular live single CD45$^+$CD11b$^+$ myeloid cells from the testes of 2-, 5- and 14-week-old wild-type (WT) mice for the indicated markers. The scale bar indicates the expression level of a given antigen from low (blue) to high (red). **c** Unsupervised FlowSOM analyses of resident live single CD45$^+$CD11b$^+$ myeloid cells in the WT testis at the indicated timepoints. The individual FlowSOM metaclusters are shown in different colors. MAC macrophage, MO monocyte. **d, e** Frequencies of macrophage (MAC) and monocyte (MO) populations in the WT testis based on the unsupervised FlowSOM analyses. **f** Unsupervised hierarchical X-shift clustering (nearest neighbor) of resident CD45$^+$CD11b$^+$ myeloid cells at indicated timepoints in the WT mouse testis. The colored boxes show the location of manually gated cell populations based on the expression analyses of myeloid cell-selective markers (shown in Supplementary Fig. 4a-c). In quantifications (**d, e**), each dot represents one mouse ($n = 4$ (**d, e**) mice per age). Data are presented as mean ± SEM (*$p < 0.05$, **$p < 0.01$, ***$p < 0.001$, ****$p < 0.0001$, two-way ANOVA with Bonferroni post-hoc test). All data are from two independent experiments. Source data are provided as a Source Data file.

labeling of testicular F4/80$^{HI}$ cells at birth, and of CD206$^+$ MHCII$^-$ macrophages in 5-week-old mice (Fig. 4c, d). Notably, yolk-sac-derived macrophages still persisted in testis (among CD206$^+$MHCII$^-$ macrophages in particular) at the age of 10 weeks (Fig. 4e). It should be noticed that the percentage of the YFP$^+$ cells in both reporter strains represents an underestimate of the real cell frequencies—as in all lineage tracing studies—due to incomplete conversion efficacy. However, when the yolk-sac-derived macrophages were depleted from the fetuses by a CD115 antibody injection at E6.5[37], we observed no decrease in the frequency or the absolute cell numbers (Fig. 4f; Supplementary Fig. 6a) in any of the main three macrophage populations in 5-week-old mice. Collectively, these data indicate that yolk-sac-derived macrophages do persist in all postnatal macrophage populations in the testis, but that they are not a prerequisite to the postnatal emergence of any of these subpopulations.

Fetal liver-derived monocytes generate a second wave of macrophage seeding during embryogenesis[8,44]. Since no fate mapping model allows specific labeling of this population, we analyzed testicular macrophage population in *Plvap*$^{-/-}$ mice, which manifest reduced numbers of fetal liver-derived macrophages in many tissues in the presence of normal production of yolk sac and bone-marrow-derived macrophages[39,45,46]. We observed a substantial reduction in the frequency of CD206$^+$MHC II$^-$ testicular macrophages in 5-week-old *Plvap*$^{-/-}$ mice (Fig. 4g). When the absolute cell numbers were analyzed, an 80% reduction in CD206$^+$MHC II$^-$ and a 50% reduction in CD206$^-$MHC II$^-$ cells were observed in testes of *Plvap*$^{-/-}$ mice when compared to littermate controls (Supplementary Fig. 6b). The absolute cell numbers of CD206$^-$MHC II$^+$ cells, in contrast, were comparable between the two genotypes. Hence, these data suggest that fetal liver-derived macrophages contribute both to CD206$^+$MHC II$^-$ and CD206$^-$MHC II$^-$ testicular macrophage populations in the 5-week-old testis.

To evaluate the contribution of postnatal, bone marrow monocyte-derived macrophages to testicular macrophage populations, we used *Ccr2*$^{-/-}$ and *Nur77*$^{-/-}$ mice. These animals manifest with sharply reduced numbers of circulating classical (Ly6C$^{Hi}$) and patrolling (Ly6C$^{Low}$) monocytes, and with severe defects in the infiltration of bone-marrow-derived macrophages in several organ systems[47,48]. However, the frequencies and absolute cell numbers of all three testicular macrophage (CD206$^+$MHC II$^-$, CD206$^-$MHC II$^-$, and CD206$^-$MHC II$^+$) populations were largely unaffected both in 5-week-old *Ccr2*$^{-/-}$ and *Nur77*$^{-/-}$ mice (Fig. 4h, i; Supplementary Fig. 6c, d). Whole-mount immunostainings of isolated seminiferous tubule segments revealed that also the localization and density of elongated F4/80$^+$ macrophages on the tubules were unaffected both in *Ccr2*$^{-/-}$ and *Nur77*$^{-/-}$ mice (Fig. 4j, k). Thus, these data strongly imply that classical CCR2-dependent infiltration of bone-marrow-derived monocytes does not significantly contribute to the testicular macrophage pool under homeostatic conditions.

**Bone-marrow-derived macrophages do not infiltrate testis.** To further address the contribution of bone-marrow-derived monocytes to the macrophage infiltration in the testis, we ablated the pre-existing tissue-resident macrophages to allow the new incoming monocytes to populate an empty niche without competition. We used an established depletion protocol of three alternate cycles of colony-stimulating factor 1 (αCSF1) antibody and clodronate-liposome (Clod) injections[9,49] (Fig. 5a). When these treatments were started in 10-day-old mice, practically all testicular macrophages had disappeared in samples taken 40 h after the completion of the last treatment (Fig. 5b, c; Supplementary Fig. 7a). When the testes were analyzed after an 11-day and 3-week recovery periods, none of the three macrophage populations were re-established when compared to control-treated (isotype antibody control + empty liposomes) mice (Fig. 5c; Supplementary Fig. 7b–e). As additional controls, we verified rapid re-establishment of blood Ly6C$^+$ monocytes in 40 h, complete recovery of bone-marrow-derived F4/80$^{Int}$ cells and the persistent absence of fetal-derived F4/80$^{Hi}$ macrophages in the kidney after the 11d recovery time (Supplementary Fig. 7f–i; for the gating strategy see Supplementary Fig. 1c, e). Hence, these depletion experiments imply that, in contrast to kidney and other organs[49], the bone-marrow-derived monocytes are not, at least in a 3-week time frame, able to populate even an empty testicular niche in pubertal mice.

Clodronate liposomes can cause local and systemic inflammatory reactions[50,51], and due to technical reasons, they cannot be repeatedly injected to newborns. We discovered that three injections of αCSF1 antibody alone (started in 10-day-old mice) caused an efficient and prolonged depletion of all testicular macrophages (Supplementary Fig. 8a, b). Notably, even a single injection of the CSF1 antibody in newborns was sufficient to cause the complete disappearance of CD206$^+$MHC II$^-$ macrophages and a significant decrease of CD206$^-$MHC II$^-$ macrophages in the testis 1 week later (Fig. 5d–g; Supplementary Fig. 9a, for the gating strategy see Supplementary Fig. 1d). After a 5-week recovery, we saw only a partial replenishment of CD206$^+$MHC II$^-$ macrophage population (Fig. 5h, Supplementary Fig 9b). In contrast, in the kidney and spleen, the macrophages were fully recovered in 5-week-old mice (Supplementary Fig. 9c). Moreover, the Ly6C$^+$ monocytes in blood and bone marrow were completely normal after a 7-day and a 5-week recovery time (Supplementary Fig 9d, e; for the gating strategy see Supplementary Fig. 1e). However, we found no evidence of increased Ly6C$^+$ monocyte infiltration in the testis of CSF1 antibody-treated mice (the numbers of testicular monocytes were on average 50.5%, 76.9%, and 65.5% from those of control-treated mice after a 24-h, 7-day, and 5-week recovery, respectively). Immunohistochemical analysis for the testis tissue sections revealed that the re-emerged CD206$^+$MHC II$^-$ cells had taken their normal position in the interstitial tissue and that they did not localize to the empty peritubular niche usually occupied by CD206$^-$MHC II$^+$ cells, which were absent following the treatment (Supplementary Fig. 9f). We also

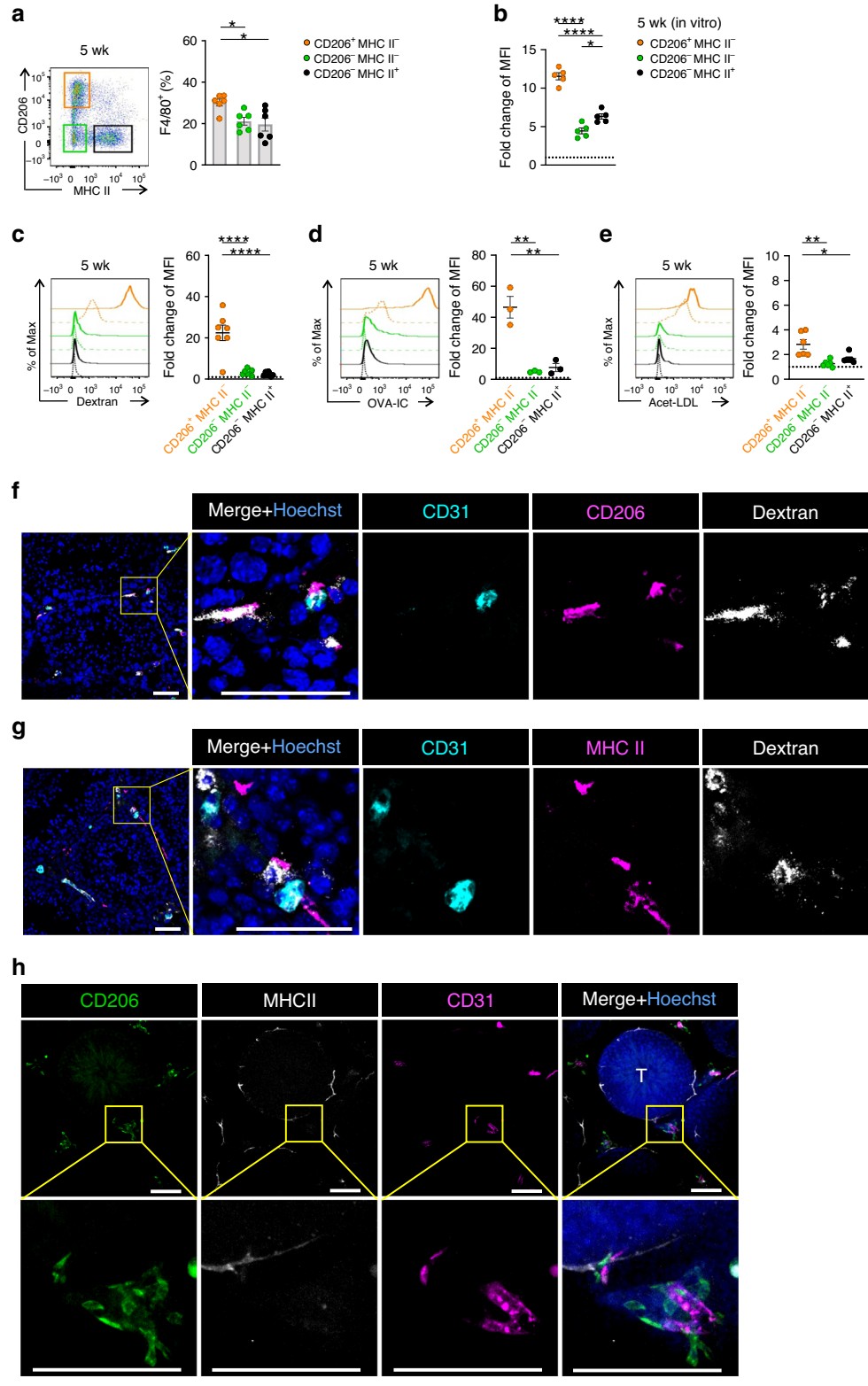

analyzed cell proliferation at the 5-week recovery timepoint using in vivo BrdU-labelings (Fig. 5i; Supplementary Fig. 9g). We found that after mock depletion, the majority of BrdU$^+$ cells fell into CD206$^+$MHC II$^-$ and CD206$^-$MHC II$^+$ populations, although some proliferation was also seen in CD206$^-$MHC II$^-$ cells. In CSF1 antibody-treated mice, in contrast, BrdU$^+$ cells were found almost exclusively among CD206$^+$MHC II$^-$ cells (Fig. 5i). Finally, after an extensive 12-week recovery period the numbers

of both CD206$^+$MHC II$^-$ cells and CD206$^-$MHC II$^+$ cells were increasing and had reached about 60% of the control values (Fig. 5e, j). Notably, CSF1 antibody injections to newborn $Ccr2^{-/-}$ mice showed that the repopulation of CD206$^+$MHC II$^-$ macrophages at 5 and 12 week was not dependent on CCR2-mediated migratory events (Supplementary Fig. 9h, i). Collectively these findings imply that after neonatal macrophage depletion, monocyte infiltration into testis is not increased.

**Fig. 3 CD206+ macrophages display superior scavenging capacity in testis. a** Fluorimetric flow analyses of CD206+MHC II− (orange gate), CD206−MHC II− (green gate), and CD206−MHC II+ (black gate) testicular macrophages of 5-week-old wild-type (WT) mice. **b** In vitro engulfment of 500 kDa fluorescent dextran at 37 °C by different macrophage subtypes isolated from the testis of 5-week-old WT mice. Baseline mean fluorescence intensity (MFI) is defined by the background binding at 4 °C. **c–e** Uptake of intravenously administered fluorescent **c** 500 kDa dextran, **d** ovalbumin (OVA)-OVA antibody complex (OVA-IC), and **e** acetylated low-density lipoprotein (acet-LDL) by testicular macrophage populations in 5-week-old WT mice (solid lines). Control mice were administered with PBS vehicle (dotted lines). Shown are representative histograms and fold change of geometric MFI in particle versus vehicle-injected mice. **f**, **g** Microscopic analyses of CD206 and MHC II with CD31 and dextran in frozen sections of testes from mice injected intravenously with fluorescent dextran. **h**, Maximum intensity projections of vibratome sections of the testis from 5-week-old WT mouse stained for CD206, MHC II, and CD31 expression. Inserts are higher magnifications from the boxed areas. T, seminiferous tubule. Scale bars (**f–h**) 50 μm. Shown are representative images from 2 (**f**, **g**) and 3 (**h**) mice. In quantifications (**a–e**), each dot represents one mouse (n = 3 (**d**), 5 (**b**), 6 (**a**, **e**), or 7(**c**) mice). Data are presented as mean ± SEM (*p < 0.05, **p < 0.01, ***p < 0.001, ****p < 0.0001, one-way ANOVA with Bonferroni post-hoc test). All data are from 2–3 independent experiments (except **b**, which is from five mice in one experiment). Source data are provided as a Source Data file.

Instead, it is likely that proliferation of the few remaining interstitial CD206+MHC II− cells (and possibly CD206−MHCII− cells) first reconstitutes the CD206+MHC II− macrophage population. These cells then appear to slowly differentiate into CD206− MHC II+ cells within 3 months.

**Macrophages are not needed for spermatogenesis postnatally.** We then studied the functional importance of monocytes and testicular macrophages for normal spermatogenesis. We found that at 5 weeks of age the relative testis weight in Ccr2−/− mice, wild-type mice treated with CSF1 antibody at birth and Ccr2−/− mice treated with CSF1 antibody at birth were similar to the control mice (Fig. 6a). Histological stainings and quantitative analyses of the testis showed an unaltered tubular morphology and diameter (Fig. 6b, c; Supplementary Fig 10a). Despite the fact that the antibody-treated mice lacked about half of interstitial macrophages and practically all peritubular macrophages (Fig. 5), they hosted normal spermatogenesis, containing a similar cellular density with full cohorts of differentiating germ cells in comparison to the control mice (Fig. 6c). In-line with these observations, histological stainings of caput epididymides showed normal presence of sperm in these mice (Supplementary Fig. 10b). Moreover, the numbers of Sertoli cells and Leydig cells were not affected either by CCR2-deficiency or by CSF1 antibody-mediated macrophage depletion (Fig. 6d–f; Supplementary Fig. 11a–f). The blood–testis barrier, a sign of immunological integrity within the testis, was also intact in all these mice (shown for CCR2-deficient mice and wild-type mice treated with CSF1 antibody in Supplementary Fig. 12a–b). Both maturing spermatocytes and spermatids were identifiable in CCR2-deficient and macrophage-depleted mice (shown for CCR2-deficient mice and wild-type mice treated with CSF1 antibody in Supplementary Fig. 12a–b). Collectively, these analyses show that CCR2-dependent bone marrow monocytopoiesis or CCR2-dependent monocyte migration to peripheral tissues are not needed for normal spermatogenesis. The macrophage depletion analyses also strongly suggest that substantial postnatal reduction of tissue-resident macrophages has no adverse effect on the normal spermatogenesis in young adult mice.

**Life-long lack of macrophages perturbs spermatogenesis.** To finally address whether macrophages at any stage of testicular development regulate male fertility, we searched for models, in which both the fetal and postnatal macrophage production would be prevented. Since CSF1 is the canonical macrophage growth factor, we first used Csf1op mice, which have a spontaneous mutation in CSF1 gene[52,53]. When analyzed at 5-week of age, we indeed found a complete lack of all three macrophage sub-populations in the testis of Csf1op mice (Fig. 7a). As an alternative,

we developed a new pharmacological approach for depleting both yolk-sac-derived macrophages (by injecting CD115 antibody at E6.5) and postnatal tissue-resident macrophages (by injecting CSF1 antibody at birth and again at the age of 2 week) in wild-type mice (Fig.7b). When analyzed after a 3-week recovery time at the age of 5 weeks, we observed a complete lack of CD206+MHC II−, CD206−MHC II−, and CD206−MHC II+ macrophages in the testis (Fig. 7c).

Analyses of Csf1op and CD115 + CSF1 antibody-injected mice revealed small reductions in the relative weight of testis or the diameter of the seminiferous tubules when compared to control mice (Fig. 7d, e). The histological analysis of the testes revealed that both models have impaired spermatogenesis (Fig. 7f). Whereas control mice contained all cohorts of differentiating germ cells, the seminiferous epithelium of the CSF1op and CD115 + CSF1 antibody-injected mice testes lacked the postmeiotic cells in nearly all seminiferous tubule cross-sections. Rare elongating spermatids were detected in the testes of CD115 + CSF1 antibody-injected mice (red arrows, Fig. 7f) and none in the CSF1op mice. However, the numbers of Sertoli and Leydig cells were not reduced by the life-long lack of macrophages (Fig. 7g, h; Supplementary Fig. 11g–j). In contrast, the blood–testis barrier appeared disorganized, and the distinct seminiferous tubule epithelial stages were not identifiable due to the absence of postmeiotic cells in both Csf1op and CD115 + CSF1 antibody-treated mice (Fig. 7i). Consistent with this, very few mature sperm cells were identifiable in the caput epididymis (Supplementary Fig. 10b) in these mice. Together with the findings of essentially normal spermatogenesis in mice lacking postnatal macrophages, the observations strongly imply that the prenatal presence of macrophages is involved in the regulation of normal spermatogenesis at puberty.

## Discussion

We report here that high-dimensional single-cell analyses reveal the presence of many more monocyte and macrophage types in the testis than previously appreciated. We also define CD206 and MHC II as superior markers to study the kinetics of three major evolving macrophage populations in the developing testis. We further show that fetal macrophages persist in the postnatal testis and that it is unlikely that bone-marrow-derived monocytes would significantly contribute to the generation of macrophages in the adult testis. Our analyses suggest that fetal-derived CD206−MHC II− macrophages originating both directly from yolk sac macrophages and fetal liver-derived monocytes will sequentially give rise to perivascular CD206+MHC II− and peritubular CD206−MHC II+ macrophages. Finally, macrophage depletion experiments reveal that prenatal macrophages control normal spermatogenesis in young adult mice, while the presence of macrophages postnatally is dispensable for this process. Collectively these data reveal hitherto unknown diversity of testicular

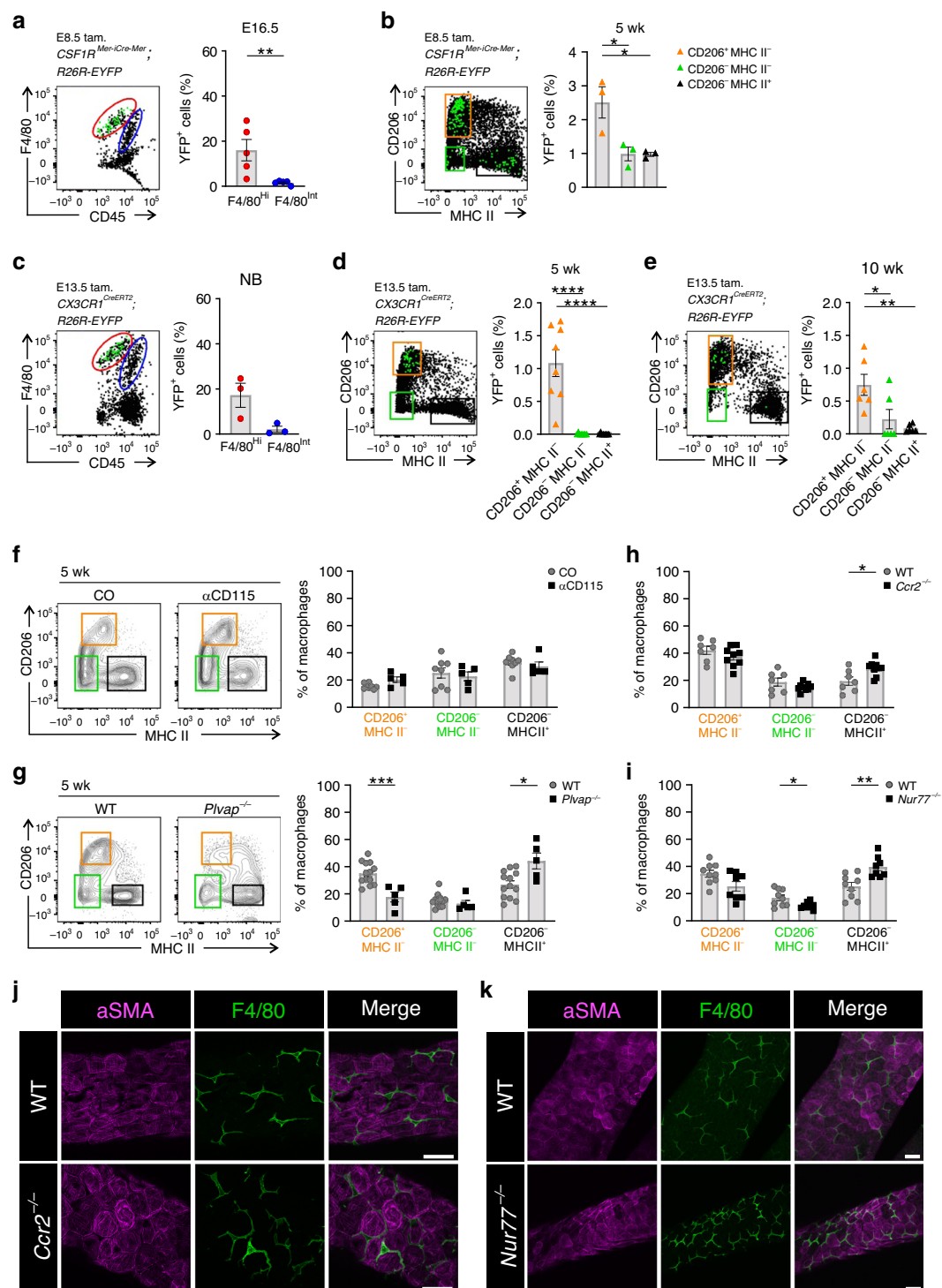

**Fig. 4 Fetal-derived macrophages dominate in the postnatal testis. a–e** Representative FACS plots and quantification of yolk-sac-derived macrophages in cell lineage mapping studies. The plots show back-gating of the YFP+ cells (green) in 5-week-old mice on the different testicular macrophage populations. **a**, **b** *CSF1R^Mer−iCre−Mer;R26R-EYFP* mice induced with tamoxifen at E8.5 and analyzed at **a** E16.5 and **b** 5 week. **c–e** *CX3CR1^CreERT2;R26R-EYFP* reporter mice induced with tamoxifen at E13.5 and analyzed at **c** birth (NB), **d** 5-week, and **e** 10-week. The quantifications show the frequency of YFP+ cells in CD206+MHC II− (orange gate), CD206−MHC II− (green gate), and CD206−MHC II+ (black gate) testicular macrophage populations. **f** Analysis of the testicular macrophage frequencies in 5-week-old WT mice treated with a depleting CD115 or control antibody at E6.5. **g–i** Flow cytometric analyses and quantifications of macrophage populations in the testis of 5-week-old (**g**) *Plvap^−/−*, (**h**) *Ccr2^−/−* and (**i**) *Nur77^−/−* mice. **j**, **k** Seminiferous tubule whole-mount stainings of WT, *Ccr2^−/−*, and *Nur77^−/−* mice with antibodies against αSMA (expressed in peritubular myoid cells), and F4/80 (a pan-macrophage marker). Shown are representative images from three mice. Scale bar (**j**, **k**) 50 µm. In the quantifications, each dot represents one mouse (*n* = 3 (**b**, **c**), 5 (**a**: YFP, **f**: CD115, **g**: *Plvap^−/−*), 6 (**e**), 7 (**h**: WT), 8 (**d**: YFP, **f**: CO, **i**: *Nur77^−/−*), 9 (**h**: *Ccr2^−/−*, **i**: WT), or 13 (**g**: WT) mice. Data are presented as mean ± SEM (*$p < 0.05$, **$p < 0.01$, ***$p < 0.001$, ****$p < 0.0001$, one-way ANOVA with Bonferroni post-hoc test (**b**, **d**, **e**) and Two-tailed Mann–Whitney U test (**a**, **c**, **g–i**)). All data are from 2–5 independent experiments. Source data are provided as a Source Data file.

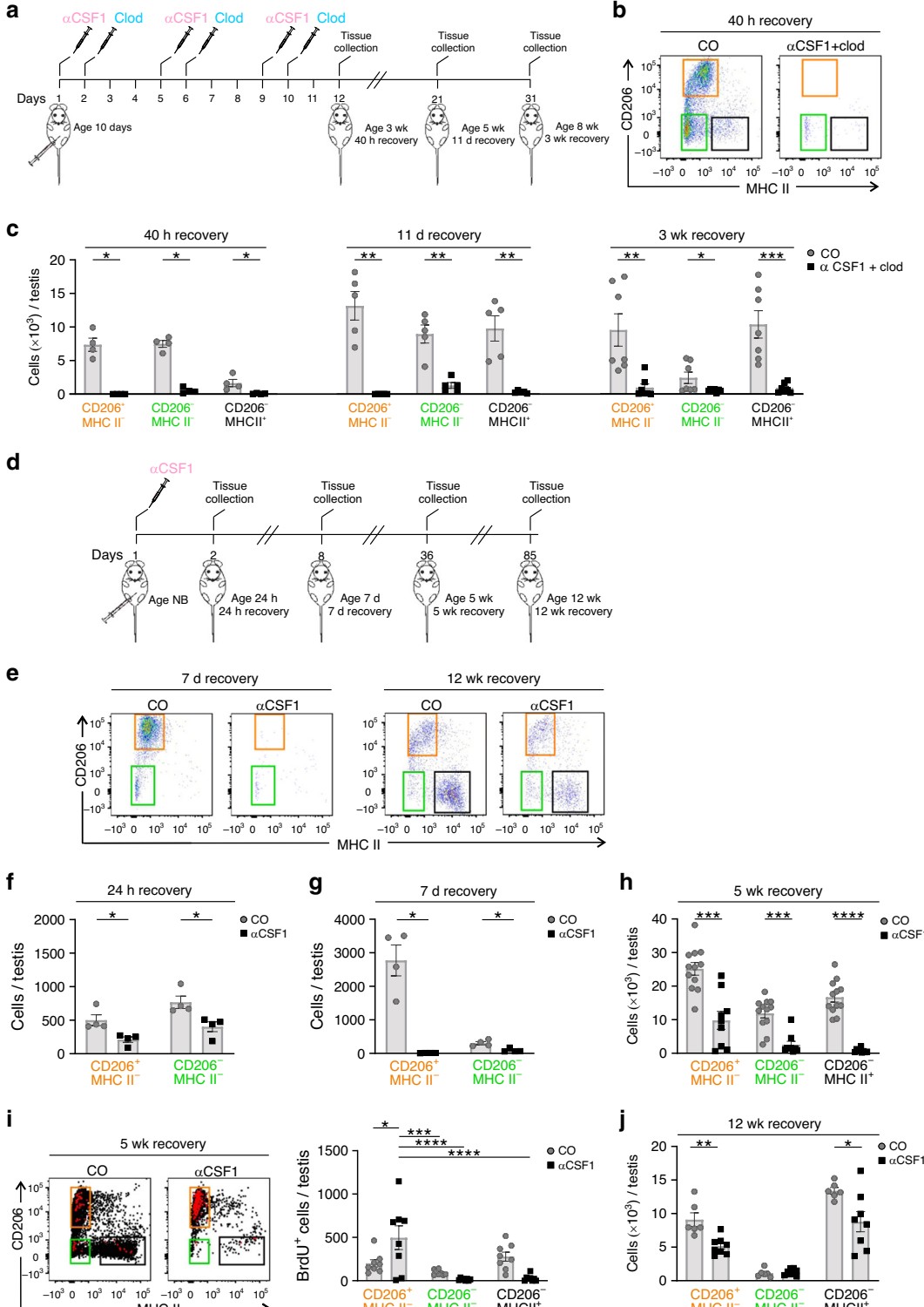

macrophages, emphasize the critical contribution of fetal-derived macrophages to the adult macrophage pool and challenge the role of postnatal macrophages in supporting gametogenesis under normal conditions.

We found that the macrophages infiltrating the fetal testis are both of yolk sac and non-yolk sac origins. In contrast, previously published data were inferred to mean that gonadal macrophages are exclusively derived from yolk sac progenitors[23]. While our staining, depletion, and fate-mapping experiments also revealed a definitive yolk-sac-derived macrophage population in the E14.5-

E17.5 testis, we identified a significant F4/80^Int population already at this timepoint. The CD115^−CD206^−Ly6C^Hi phenotype of these cells was strikingly different from the CD115^+CD206^+ Ly6C^−/Low phenotype of the yolk-sac-derived cells, and strongly suggested that they originated from fetal liver monocytes. The resistance of F4/80^Int cells against depletion of yolk-sac-derived cells and their lack of labeling in CSF1R- and CX3CR1 cell-fate mapping experiments verified their non-yolk sac origin. The discrepancy between the findings of our study and a previous study is readily explained by the fact that only converted cells and

**Fig. 5 Bone-marrow-derived monocytes do not repopulate an empty testicular niche in postnatal mice. a** Experimental design for depleting macrophages with CSF1 antibody (αCSF1) and clodronate (Clod) liposome treatment. Control mice (CO) were treated with isotype-matched control antibody and empty liposomes at the same timepoints. The age of the mice and the recovery times after the last injection are also indicated. **b** A representative FACS plot of testis after a 40 h recovery. **c** Total numbers of CD206$^+$MHC II$^-$ (orange gates), CD206$^-$MHC II$^-$ (green gates), and CD206$^-$MHC II$^+$ (black gates) testicular macrophage populations after a 40 h, 11-day, and 3-week recovery period (from **a**). **d** Experimental design for depleting macrophages with a single injection of CSF1 (αCSF1) or control (CO) antibody at birth. **e** Representative FACS plots after a 7-day and 12-week recovery (from **d**). **f–j** Total numbers of testicular macrophages (**f–h**, **j**) after a 24-h, 7-day, 5-week, and 12-week recovery from the injection (from **d**). **i** In vivo BrdU-labeling of the recovering testicular macrophages at the 5-week timepoint. BrdU$^+$ cells (red) are back-gated on the different testicular macrophage populations. In the quantifications, each dot represents one mouse ($n = 4$ (**c**: 40 h;CO,αCSF1 + clod, **f**: 24 h, **g**: 7d per treatment), 5 (**c**: 11d; CO,αCSF1 + clod), 6 (**j**: CO), 7 (**c**:3 week;CO,αCSF1 + clod), 8 (**i**: CO,αCSF1, **j**: αCSF1),9 (**h**: αCSF1), or 12(**h**: CO) mice). Data are presented as mean ± SEM (*$p < 0.05$, **$p < 0.01$, ***$p < 0.001$, ****$p < 0.0001$, Two-way ANOVA with Bonferroni post-hoc test (**i**) and Two-tailed Mann–Whitney U test (**c**, **f–h**, **j**)). Data are from 2–3 independent experiments. Source data are provided as a Source Data file.

only E14.5 and earlier timepoints were included in the analyses in the previous study[23], which made it impossible to detect the fetal liver-derived macrophages. Thus, our results for the first time reveal a dominant contribution of fetal liver-derived cells to the macrophage pool of fetal testes.

The peritubular macrophages in the testis have been proposed to be completely derived from bone-marrow-derived cells and first appear during the prepubertal period[31]. Similarly, bone-marrow-derived cells have also been reported to increasingly supersede the fetal-derived cells among the interstitial macrophages during the postnatal testis development[31]. These conclusions were obtained from phenotypic analyses of cell populations from which Ly6C$^+$ cells were gated out and CD64 was used as a discriminative marker between peritubular and interstitial cells[31].

Our data, in contrast, show no significant contribution from bone-marrow-derived cells, and these cells were even unable to fill an empty testicular niche postnatally. When comparing the results it is important to note that we demonstrate a significant contribution of Ly6C$^+$ cells to the testis macrophages in newborns and the ambiguity of CD64 as a marker for macrophage subpopulations. Furthermore, the presence of a major CD206$^-$ MHC II$^-$ population went unnoticed in the previous work due to overlaid gating[31]. In our analyses, this CD206$^-$MHC II$^-$ population, but not monocytes, had high proliferative activity and clear trajectories in newborn and young mice to the CD206$^+$MHC II$^-$ macrophage population and later to the CD206$^-$MHC II$^+$ peritubular population. In addition, we found that testicular macrophages were completely intact in Ccr2$^{-/-}$ mice, which postnatally show a severe reduction of monocyte-dependent macrophage generation in many organs[47]. Notably, fertility is also normal in Ccr2$^{-/-}$ mice[47].

When we experimentally depleted macrophages in postnatal mice, they rapidly reappeared in the kidney and spleen upon the re-establishment of bone marrow monocytopoiesis. In the testis, in contrast, neither a macrophage recovery with similar kinetics nor an increase in monocyte infiltration was observed. The slow and partial recovery of CD206$^+$MHC II$^-$ and CD206$^-$MHC II$^-$ cells before that of CD206$^-$MHC II$^+$ peritubular cells—the previously claimed bone-marrow-derived cells—suggests that the few depletion-resistant fetal-derived macrophages play an important role in the recovery. We also show for the first time that macrophages are in contact with seminiferous tubules already at birth. Moreover in 1-week-old mice, these macrophages display the elongated, branching morphology along the tubules but still do not express MHC II, which was previously used for their identification[31]. Collectively, our data using high-dimensional phenotyping, in silico and in vivo fate mapping, knockout mice, depletion experiments, and intraorgan localization analyses strongly imply that CD206$^-$MHC II$^+$ peritubular macrophages represent fetal cell types, which undergo extensive

phenotypic maturation postnatally rather than newly generated descendants of bone-marrow-derived monocytes.

It has been previously shown that testicular macrophages support testosterone production in vitro, that local clodronate injections abolish Leydig cells, and that macrophage-impaired Csf1$^{op}$ mice are subfertile[54,55]. These old findings have laid the basis for the current paradigm that testicular macrophages are important for reproduction. However, more recent data indicate that macrophage depletion by CD11b-diphtheria toxin model only reduces intratesticular testosterone levels by 50% and has no impact on blood testosterone levels or Sertoli cells[29]. Only the number of spermatogonial precursors was reduced in the macrophage-depleted mice, possibly due to diminished CSF1 and retinoic acid synthesis by resident macrophages. In our experiments, we found no impairment of spermatogenesis in the testis practically lacking macrophages postnatally. By detailed analysis of genetic and pharmacological models, in which macrophages are missing throughout the whole life span of the mouse, we found severe impairment of spermatogenesis in young adult mice. Our data thus supports the concept that macrophages serve important reproductive functions prenatally, possibly in the testis morphogenesis[23], but that their presence after the birth is not needed for normal spermatogenesis.

In conclusion, our results alter many previously established concepts in the macrophage biology of the testis. Our data show that fetal liver-derived macrophages rather than yolk-sac-derived macrophages or bone-marrow-derived cells are the dominant cells among both interstitial and peritubular macrophages, and that macrophages are present at both testicular compartments already at birth. Our in silico fate mapping analyses tentatively identify fetal-derived CD206$^-$MHC II$^-$ macrophages as precursor cells, which sequentially give rise to both interstitial CD206$^+$MHC II$^-$ and peritubular CD206$^-$ MHC II$^+$ macrophages under physiological conditions. When repopulating an empty testicular niche postnatally, locally proliferating CD206$^+$MHC II$^-$ cells likely differentiate into CD206$^-$MHC II$^+$, which then take their peritubular position. The formal experimental validation of these developmental pathways will require the development of new fate-mapping mouse lines especially for tracing fetal liver-derived macrophages. Our data together with other recent research[29] also argue against a prerequisite role of postnatal testicular macrophages in spermatogenesis. Hence, our high-dimensional single-cell analyses of testicular macrophages will be useful for the re-examination of the role of macrophages in prenatal testicular development and in immune functions during normal and pathological spermatogenesis.

## Methods
**Mice**. Plvap$^{tm1Salm}$ mice (referred as Plvap$^{-/-}$) have been described previously[39,46]. Ccr2$^{-/-}$ (stock 004999), Nur77$^{-/-}$ (stock 006187), R26R-EYFP

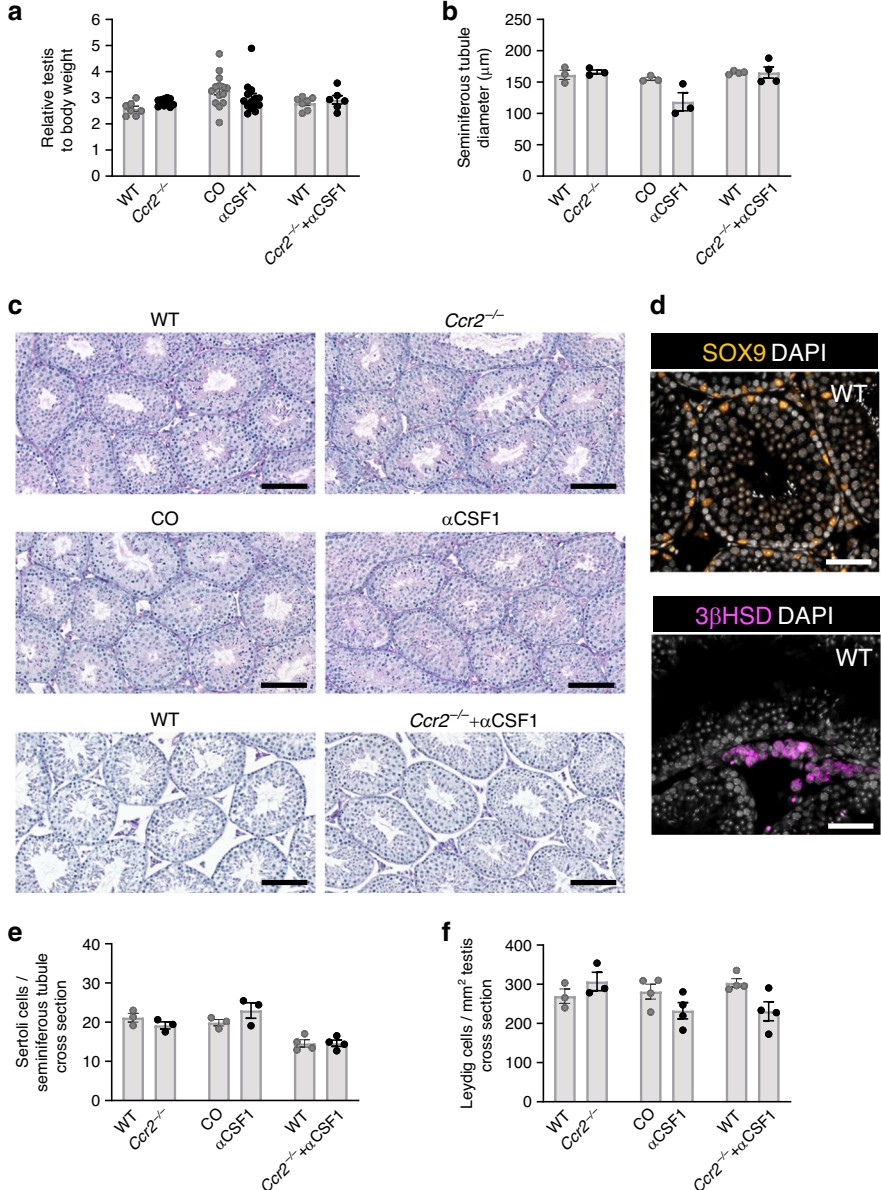

**Fig. 6 Spermatogenesis is independent of monocytes or macrophages postnatally. a–f** Analyses of testis in 5-week-old *Ccr2*−/− mice, wild-type (WT) mice treated with CSF1 antibody at birth (αCSF1), and *Ccr2*−/− mice treated with CSF1 antibody at birth (*Ccr2*−/−+αCSF1), and in their appropriate controls (WT, wild-type mice; CO, isotype-matched control antibody-treated mice). **a** Relative weight of testis (testis weight (mg)/body weight (g)). **b** Diameter of the seminiferous tubule. **c** Histology of testis (PAS stainings). **d** Representative images of immunohistological stainings for Sertoli cells (SOX9) and Leydig cells (3βHSD), white is DAPI. **e** Numbers of Sertoli cells per seminiferous tubule cross-section. **f** Numbers of Leydig cells per cross-sectional testicular area. Shown are representative images of three mice. Scale bars **c** 100 μm, **d** 50 μm. In the quantifications, each dot represents one mouse(*n* = 3 (**b**, **e**, **f:** WT, *Ccr2*−/−, CO, αCSF1), 4 (**b**, **e**, **f:** WT,: *Ccr2*−/−+αCSF1), 6 (**a:** *Ccr2*−/−+αCSF1), 7 (**a:**WT), 10 (**a:** *Ccr2*−/−), or 14(**a:**CO, αCSF1) mice). Data are presented as mean ± SEM. All data are from 2–3 independent experiments. Source data are provided as a Source Data file.

(stock 006148), *CX3CR1*^CreERT2 (stock 020940), *Csf1r*^Mer-iCre-Mer (stock 019098), and *Csf1*^op (stock 000231) mice were obtained from The Jackson Laboratory. Wild-type C57BL/6 J and C57BL/6 N mice were purchased from Janvier Labs. Mice were bred and housed under controlled environmental conditions (12 h light/12 h darkness, temperature range of 20–23 °C, relative humidity to 50–60%) and under specific pathogen-free conditions at the Central Animal Laboratory of the University of Turku. Animal experiments were approved by the National Animal Experiment Board in Finland (Animal license number 6211/04.10.07/2017). They were carried out in adherence to the rules and regulations of the Finnish Act on Animal Experimentation (497/2013), which is fully comparable to the U.S. National Institutes of Health guidelines on animal experimentation, in accordance to the 3R-principle. Age and sex-matched wild-type mice were used as controls in each experiment. Gestational age was estimated considering the day of appearance of the vaginal plug as embryonic day 0.5 (E0.5).

**Timed pregnancies and in utero tamoxifen administration.** To study fetal-derived macrophages *CX3CR1*^CreERT2 and *Csf1r*^Mer-iCre-Mer mice were crossed with *R26R-EYFP* reporter mice. For tamoxifen induction of CRE activity, a single dose of tamoxifen (1.5 mg, mended with 0.75 mg progesterone; Sigma–Aldrich) was administered intraperitoneally to pregnant females at the timepoints indicated in the Figures. This induction scheme leads to selective labeling of yolk-sac-derived macrophages at E8.5 timepoints in the CSF1R-reporter mice, since the other macrophage types have not yet emerged[10]. Also, the E13.5 induction of CX3CR1-reporters labels yolk-sac-derived macrophages only since CX3CR1 is not expressed in fetal liver-derived monocytes or their precursors[14,32].

**Macrophage depletion.** To deplete yolk-sac-derived macrophages, pregnant females were treated with a single intraperitoneal injection of 3 mg of CD115 blocking antibody (Bio X Cell, clone AFS98) or IgG2a isotype control (Bio X Cell,

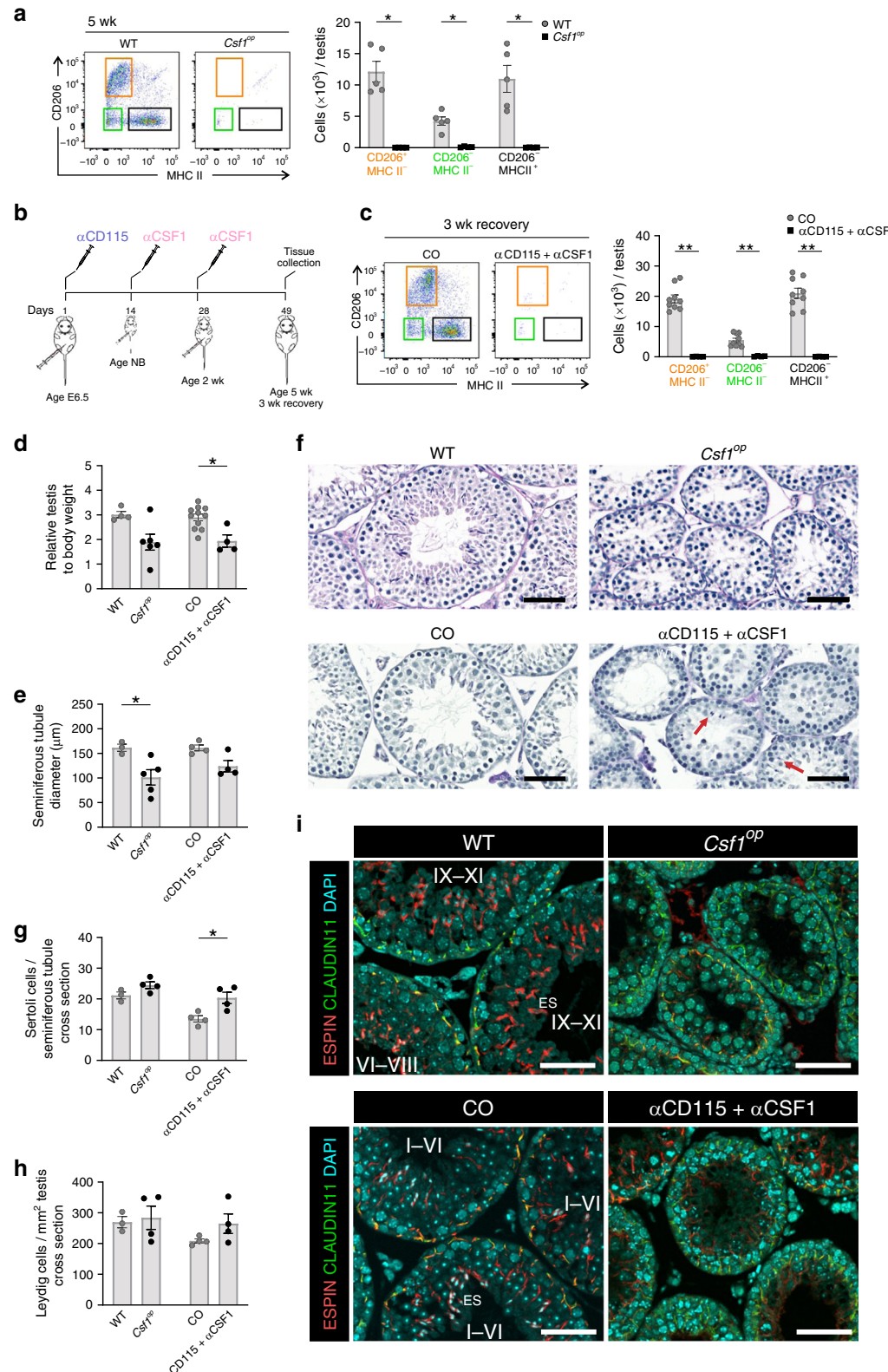

clone 2A3) at E6.5 as described previously[9,37,38]. Tissues were harvested at E17.5 or postnatal age of 5 weeks for flow-cytometry analyses.

Ten-day-old wild-type mice were treated with CSF1 neutralizing antibody and clodronate-containing liposomes to deplete tissue macrophages and blood monocytes. Intraperitoneal injection of CSF1 antibody (Bio X Cell, clone 5A1) or IgG1 isotype control (Bio X Cell, clone HRPN) (500 μg on the first cycle, 250 μg on the second and third cycles) was followed by intravenous administration of clodronate-containing or empty control liposomes (Liposoma; in 50 μl volume) on a subsequent day. Three consecutive treatment cycles were performed at three-day

intervals. Tissues were harvested for flow cytometry and imaging analyses 40 h, 11 days, or 21 days after the last clodronate injection (Fig. 5a).

To deplete tissue macrophages from newborn wild-type mice, a single 150 μg dose of neutralizing CSF1 antibody (Bio X Cell, clone 5A1) or IgG1 isotype control (Bio X Cell, clone HRPN) was administered intraperitoneally at postnatal days 0–2 to wild-type and $Ccr2^{-/-}$ mice. Tissues were harvested for flow-cytometry and imaging analyses after 24 h, 7 days, 5, or 12 weeks of recovery (Fig. 5d).

To deplete all yolk-sac-derived macrophages already during the development and all tissue-resident macrophages postnatally we developed a new combination

**Fig. 7 Prenatal macrophages support spermatogenesis. a** FACS analyses of testicular macrophages in 5-week-old *Csf1op* mice. **b** The experimental protocol for sequential treatment of mice with CD115 and αCSF1 antibodies. **c** Analyses of the testicular macrophages in 5-week-old mice after the pharmacological depletion of pre and postnatal macrophages with CD115 + αCSF1 antibodies. **d–h** Analyses of relative testis weight (**d**), seminiferous tubule diameter (**e**), testicular morphology (**f**), Sertoli cell numbers (**g**), and Leydig cell numbers (**h**). **i** Representative figures of blood–testis barrier immunostained for CLAUDIN 11 (to detect tight junctions between Sertoli cells) and ESPIN (to detect actin bundles on the Sertoli cell-side of the basal and apical ectoplasmic specializations) in 5-week-old mice. Specific stages of the seminiferous epithelial cycle (I–VI, VI–VIII, IX–XI) are indicated in WT and CO mice but not identifiable in *Csf1op* or in CD115 + αCSF1 mice. ES elongating spermatids. Scale bar (**f, i**) 50 μm. In the quantifications, each dot represents one mouse (*n* = 3 (**e, g, h**: WT), 4 (**c, d, e, g, h**: CD115 + αCSF1, **a, g, h**: *Csf1op*, **d**: WT, **e, g, h**: CO), 5 (**a**: WT, **e**: *Csf1op*), 6 (**d**: *Csf1op*), 9 (**c**:CO), or 11 (**d**: CO) mice). Data are presented as mean ± SEM (*\*p* < 0.05, \*\**p* < 0.01, Two-tailed Mann–Whitney U test (**a, c–e, g**). All data are from 2–3 independent experiments. Source data are provided as a Source Data file.

treatment protocol (Fig. 7b). A single 3-mg dose of CSF1R blocking antibody (Bio X Cell, clone AFS98) or IgG2a isotype control (Bio X Cell, clone 2A3) was injected intraperitoneally at E6.5. Then 150 μg of αCSF1 antibody (Bio X Cell, clone 5A1) or IgG1 isotype control (Bio X Cell, clone HRPN) was injected intraperitoneally to newborn males, and for a second time at postnatal days 14–15 with a dosage of 500 μg. Tissues were harvested for flow cytometry and imaging analyses after a 3-week recovery at the age of 5 weeks.

**Cell isolation.** The pregnant females were euthanised at day 14.5 or 16.5 post coitum by carbon dioxide ($CO_2$) asphyxiation and cervical dislocation. Embryos were dissected from the uterus and euthanised by decapitation. Newborn and 7-day-old pups were euthanized by decapitation and from 2 week of ages onwards mice were euthanized by $CO_2$ asphyxiation and subsequent cervical dislocation or cardiac puncture.

Fetal testes were dissociated to flow-cytometry buffer (2% fetal calf serum (FCS; v/v) and 0.05% $NaN_3$ (v/v) in phosphate-buffered saline (PBS)) by pipetting. Embryonic brain was minced and digested with 50 μg/ml DNase 1 (Roche, cat. 10104159001) and 1 mg/ml Collagenase D (Roche, cat. 1108886601) in Hanks' solution at +37 °C for 20 min. Leukocytes were then separated from stromal cells by discontinuous Percoll[TM] (GE Healthcare, cat. 17-0891-01) gradient centrifugation method and the microglial cells were isolated as previously described[39,56]. Kidneys and spleen of 7-day-old pups were minced and digested with 50 μg/ml DNase 1 and 1 mg/ml Collagenase D in Hanks' solution at +37 °C for 45 min.

Blood from newborn and 7-day-old mice was collected by bleeding the body to heparin-PBS (50 μl of 100 IU/ml heparin in 500 μl of PBS). The postnatal blood was drawn by cardiac puncture into heparinized syringes and erythrocytes were lysed from the blood and spleen samples as described[57].

Postnatal testes were minced and digested with 50 μg/ml DNase 1 and 1 mg/ml Collagenase D in Hanks' solution at +37 °C for 45 min. Kidneys were homogenized to RPMI-1640 medium in gentleMACS C-tube (Miltenyi Biotech) with gentleMACS Dissociator (Miltenyi Biotech) and leukocytes were isolated by Optiprep[TM] (Sigma–Aldrich, cat. D1556) gradient centrifugation[39]. The bone marrow was isolated by gently crushing the femurs. Finally, the isolated cells were washed and suspended in PBS and all cell suspensions were filtered through a silk cloth (pore size 77 μm).

**Antibodies.** All antibodies and dilutions used for flow cytometry, mass cytometry, and immunofluorescence stainings are listed in Supplementary Table 1. Unconjugated antibodies were labeled with [156]Gd, [158]Gd, or [166]Er using Maxpar Antibody Labeling Kits (Fluidigm, cat. 201156 A, 201158 A, and 201166 A, respectively) according to the manufacturer's protocol.

**Flow cytometry.** Cells were stained with Fixable Viability Dye eFluor 780 (eBioscience, cat. 65-0865) to label the dead cells. Unspecific binding to low-affinity Fc-receptors was blocked by incubating the cells with unconjugated CD16/32 antibody (Bio X Cell, clone 2.4G2). Cells were subsequently stained for 30 min at 4 °C with antibodies diluted to the FACS-buffer, washed and fixed with 1% formaldehyde in PBS. Samples were acquired with LSRFortessa flow cytometer with FACSDiVaTM version 8 software (Becton Dickinson), and data were analyzed with the FlowJo software (FlowJo LLC).

For in vivo bromodeoxyuridine labelings, 120 μl of 10 mg/ml BrdU solution (BD Bioscience) was injected i.p. to mice 2 h before the sacrifice. The testicular and bone marrow cells were isolated, stained for leukocyte markers, fixed, and finally stained with FITC-conjugated BrdU antibody (BrdU Flow kit, BD Bioscience).

The gating strategies for flow cytometric analyses are depicted in Supplementary Fig. 1a–e. When indicated, the total cell number of a given macrophage subtype per the testis was determined numerically by dividing the number of events acquired with flow cytometry by the volume fraction subjected to the flow analyses and then multiplying that value with total sample volume. If more than one testis was used for a sample, then the result was further divided by the number of the testes.

**Mass cytometry.** The cells were isolated as described above for flow cytometry. Dead cells were excluded with 2.5 μM Cell-ID Cisplatin staining (Sigma–Aldrich, cat. 479306-1 G) at RT for 5 min. Next, the cells were washed, Fc-blocked with CD16/32 antibody, and stained with a heavy-metal isotope-labelled mAb cocktail (Supplementary Table 1) for 30 min at RT. Cells were then incubated for 1 h with intercalation (Cell ID Intercalator-103Rh in MaxPar® Fix and Perm Buffer; Fluidigm, cat. 201103 A and 201067, respectively) at RT. Finally, the cells were fixed with 4% paraformaldehyde solution (PFA; Santa Cruz Biotechnology, cat. sc-281692) overnight and pelleted. Before data acquisition with a CyTOF mass cytometer, cells were resuspended to MaxPar Water (Fluidigm cat. 201069). Mass cytometry data were acquired with CyTOF 6.7 system control software (Fluidigm).

Mass cytometry data were bead-normalized and exported as flow-cytometry file (FCS) format to Cytobank (Cytobank, https://www.cytobank.org). Before downstream analysis, live (Cisplatin−), singlet ([103]Rh[Int]), CD45+CD11b+F4/80+ (newborn) or CD45+CD11b+ (2-, 5-, and 14 week old) cells were manually gated (Supplementary Fig. 1f–h). Gating, viSNE plots (dimensionality reduction algorithm t-SNE), and unsupervised clustering by FlowSOM algorithm were performed with Cytobank platform.

Unsupervised hierarchical clustering was performed using the X-shift algorithm which was run in the VorteX clustering and visualization environment (version VorteX 29-Jun-2017-rev2). For the clustering, the default settings were used with the nearest density estimation (K) from 150 to 10, with 30 steps. The force-directed layout was created from all five clusters in newborns (K = 68), 13 clusters in 2-week-old (K = 72), 19 clusters in 5-week-old (K = 108), and 19 clusters from 14-week old (K = 58) mice (ForceAtlas2 algorithm; all cell events from clusters smaller than 1000 events and 1000 randomly selected events from the clusters bigger than 1000 events). The cell clusters were manually assigned to different leukocyte subpopulations based on the expression of the cell-type-selective leukocyte differentiation markers. The layout and visualization were produced with Gephi 0.9.1 (https://gephi.org).

Manual bi-axial blotting of the mass cytometry data was used to allow direct comparison with the fluorimetric data (Supplementary Figs. 1h and 5a).

**In vivo and in vitro scavenging experiments.** Immune complexes (OVA-IC) of ovalbumin (OVA)- Atto488 (Sigma–Aldrich, cat. 41235) and rabbit polyclonal OVA IgG (Sigma–Aldrich, cat. C6534) were prepared in vitro as previously described[9]. One hundred and twenty microgram of OVA-IC (5:1 ratio of OVA and OVA antibody), 10 μg of fluorescently labeled acetylated low-density lipoprotein (LDL; Alexa Fluor 488 conjugated, Thermo Fisher Scientific, cat. L23380), or 0.8 mg of 500 kDa Dextran (fluorescein-conjugated, Thermo Fisher Scientific, cat. D7136) diluted in PBS were administered via tail vein injections to 5-week-old wild-type mice in a 150 μl volume. Control mice were injected with 150 μl PBS. The recipient mice were sacrificed after 1 h (dextran) or 2 h (LDL and OVA-IC), and the testes were collected for flow cytometry and imaging analyses. To exclude cell population-specific autofluorescence, the MFI values are represented as fold changes compared to the control by dividing the MFI value for each particle with the MFI value obtained from the control sample.

For in vitro endocytosis assays, CD11b+ cells were positively selected from single-cell suspension of testes of 5-week-old wild-type mice (digested as described in "Cell isolation") by CD11b MicroBeads system (MiltenyiBiotec, cat. 130-049-601) according to manufacturer's protocol. $3.0 \times 10^5$ cells/well were seeded to 96-well-plates and 100 μl of (0.5 mg/ml) 500 kDa fluorescein-conjugated Dextran diluted in RPMI-1640 (supplemented with 10% FCS and 2 mM L-glutamine) was added to wells. Plates were incubated at +37 °C or +4 °C (as a negative control) for 1 h. Thereafter, the cells were washed extensively with PBS and stained with antibodies (CD45, CD11b, F4/80, CD206, and MHCII) for flow cytometry. The endocytosis capacity of each cell population was calculated by dividing the MFI of samples incubated at +37 °C by the MFI of samples incubated at +4 °C.

**Histology.** Testes were fixed in Bouin's solution (Sigma–Aldrich, cat. HT10132) or 4 % PFA solution (Santa Cruz Biotechnology, cat. sc-281692) at RT for 2 h or at +4 °C overnight. Epididymides were fixed in paraformaldehyde. After serial of washes with PBS and dehydration in ethanol series of increasing concentrations, the samples were embedded in paraffin. Four micrometer thick sections were cut

for the histological analyses. Sections were deparaffinized and stained with hematoxylin and eosin or periodic acid-Schiff (PAS) and imaged using Pannoramic P250 slide scanner with the 20x objective (3DHISTECH Ltd.).

**Immunofluorescence stainings and confocal imaging**. To quantify the number of Sertoli and Leydig cells, indirect immunofluorescence was performed as described previously[58]. Briefly, antigens were retrieved from sections of PFA fixed testes with 0.1 M citrate buffer, pH 6 at 95 °C, and autofluorescence blocked by incubation in 0.1 M $NH_4Cl$. After blocking nonspecific binding with 5% normal donkey serum (NDS), samples were incubated with primary antibodies against SOX9 or 3βHSD. After incubation with secondary antibodies diluted in 5% NDS the, sections were mounted with SlowFade™ Diamond Antifade mounting medium containing DAPI (Thermo Fisher Scientific, cat. S36964). All sections were scanned with a Pannoramic Midi fluorescence slide scanner (Objective ×40, 3DHISTECH Ltd) and analyzed with CaseViewer 2.4 software. Images were also captured using LSM880 confocal microscope (Carl Zeiss) equipped with Plan-Apochromat ×20/0.8 objective and Zen 2010 software (Zeiss).

The number of Sertoli cells (SOX9-positive cells) was counted from a minimum of 30 rounded seminiferous tubule cross-sections in each testicular section. The number of Leydig cells (3βHSD-positive cells) was calculated in a measured testicular area. Blood–testis barrier integrity was analyzed at different spermatogenic stages of the seminiferous epithelium[59].

The diameter of the seminiferous tubules was measured by using the pannoramic viewer 1.15 software (3DHISTECH Ltd). Only rounded cross-sections of seminiferous tubules were chosen for quantification (Supplementary Fig. 10a). The average of two perpendicular measurements was recorded and a minimum of 30 seminiferous tubules cross-section from different regions of the testis were analyzed per testis section.

For seminiferous tubule whole-mount staining the testes of 7-days-, 2-, and 5-week-old mice were dissected and the tunica albuginea was removed. The seminiferous tubules were prepared for whole-mount stainings as previously described[60]. In brief, the samples were blocked (0.3% Triton X-100, 2% bovine serum albumin (BSA), and 10% fetal bovine serum (FBS)), and then sequentially incubated with primary unconjugated antibodies, secondary antibodies and directly conjugated antibodies diluted in appropriate blocking solutions. Finally, the seminiferous tubules were arranged in linear strips and mounted with SlowFade™ Diamond Antifade mounting medium containing DAPI (Thermo Fisher Scientific, cat. S36964). Images were captured using LSM780 confocal laser-scanning microscope (Carl Zeiss) equipped with Plan-Apochromat ×20/0.8 or C-Apochromat ×40/1.2 objective and Zen 2010 software (Zeiss).

For frozen section stainings, testes were embedded to optimal cutting temperature (OCT) compound (Tissue-Tek, cat. 4583) and frozen with liquid nitrogen. Samples were cut into 5 μm thick sections with cryo microtome, followed by immediate fixation with ice-cold 95% ethanol. Slices were sequentially incubated with primary antibodies (30 min), secondary antibodies (30 min), and directly conjugated antibodies. After staining with Hoechst (Thermo Fisher Scientific, cat. 62249) the sections were mounted with ProLong Gold without DAPI (Thermo Fisher Scientific, cat. P36930).

For whole-mount stainings, the testes were lightly fixed with 2% PFA in PBS for 1 h on ice, washed with PBS three times 10 min on ice, and dehydrated in an increasing methanol series. Samples were cooled to −20 °C in 100% methanol and subsequently rehydrated in a decreasing methanol series on ice. The testes were then embedded in 4% low-melting agarose (Lonza, cat. 50080) and the polymerized block was sectioned with VT1200 S vibratome (Leica) to 300-μm thick slices. Floating sections were isolated from agarose, blocked, incubated with primary and secondary antibodies, and then with directly conjugated antibodies as previously described[39]. After staining with Hoechst, the samples were dehydrated with increasing methanol series and subsequently optically cleared in glass-bottom microwell dishes first with 50% benzyl alcohol (Honeywell, cat. 402834) 1:2 benzyl benzoate (Sigma–Aldrich, cat. B6630) (BABB) in methanol and then with 100% BABB, which was also used as a mounting medium.

Whole-mount vibratome and frozen section samples were imaged with a spinning disk confocal microscope (Intelligent Imaging Innovations) equipped with Orca-Flash4 v2 sCMOS camera (Hamamatsu) and Plan-Apochromat ×20/0.8 or C-Apochromat ×40/1.1 objective and using SlideBook 6 software (Intelligent Imaging Innovations).

Maximum intensity projections of z-stacks acquired from vibratome sections were produced from 3–48 slices with a step size of 0.43 or 2.34 microns. Background subtraction with rolling ball method and linear brightness and contrast adjustments for these projections and images acquired from frozen sections were done with ImageJ software (http://rsb.info.nih.gov/ij).

For the quantifications of vessel-associations, cells with either CD206 or MHCII signal were enumerated and visually assigned to those in direct contact with CD31+ blood vessels and to those not in contact with blood vessels. The analyses were made from the original 16-bit Z-stacks acquired from vibratome sections with linear brightness and contrast adjustments using Image J.

**Statistical analysis**. All adult mice were assigned to experimental groups without specific randomization methods because comparisons involved mice of distinct genotypes. The investigators were blinded to the genotype of the embryos during the experimental procedures. Comparisons between the study groups were done using GraphPad Prism software v8 (GraphPad Software Inc). All data are presented as mean values ± SEM. Statistical significances between groups were determined using the Mann–Whitney U test, One-way or Two-way ANOVA with Bonferroni post-hoc test. $P$-values < 0.05 are considered to be statistically significant.

**Reporting summary**. Further information on research design is available in the Nature Research Reporting Summary linked to this article.

## Data availability

All materials used in this study are available commercially or from the authors. The sources for the materials are in the Materials and Methods section. Supplementary Information file is available in online version of this article. All raw data that support the findings of this study are available from the corresponding authors upon reasonable request. Mass cytometry data files are also available from FlowRepository (FR-FCM-Z2RQ). The source data underlying Figs. 1a, d, 2d, e, 3a–e, 4a–i, 5c, f–j, 6a, b, e, f, 7a, c–e, g, h and Supplementary Figs. 2a, g, 5a, 6a–d, 7a, c, e–i, 8b, and 9c–f, h–j are provided as a Source Data file with this paper. Source data are provided with this paper.

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

## Acknowledgements

We thank Ms. Etta-Liisa Väänänen, Laura Grönfors and Norma Jäppinen for expert technical help and The Cell Imaging and Cytometry Core facility (Turku Bioscience, University of Turku, Åbo Akademi University, and Biocenter Finland) is acknowledged for services, instrumentation, and expertise. This study was financially supported by the Academy of Finland, Sigrid Juselius Foundation, Jane and Aatos Erkko Foundation, The Cancer Foundation Finland, Emil Aaltonen Foundation, Orion Research Foundation sr and University of Turku Doctoral programme for Molecular Medicine

## Author contributions

E.L. designed experiments, analyzed data, and wrote the manuscript. L.L. performed experiments and contributed to experiment design, and analysis. S.C-M. and J.M. performed experiments, contributed to data analysis. S.T., V.O., and H.G. performed experiments. J.T. contributed to the preparation of the manuscript. P.R. and M.S. conceived and supervised the study and wrote the manuscript.

## Competing interests

The authors declare no competing interests.
