## [Peer Review File · Nature Communications]

Reviewers' comments:

Reviewer #1 (Hematopoiesis, tissue-resident macrophage)(Remarks to the Author):

Lokka et al. present data on the heterogeneity and origin of testes-resident macrophages and their role in spermatogenesis. Using high dimensional single cell mass cytometry, alterations in macrophage populations from the embryo to adult mice are determined, thereby defining new macrophage subsets. Lineage-tracing and macrophage-depletion studies substantiate the interpretation of the mass cytometry data and the manuscript elegantly bridges descriptive high-dimension mass-cytometry with functional in vivo approaches. The key messages are new and unprecedented to the field, making this a potentially high impact publication. All experiments are well thought through, presented and described. However, a few points need to be addressed to substantiate the strong and important conclusions drawn from the data provided:

- 1) The mass cytometry data is very fascinating and a lot can be learned about macrophage heterogeneity in the testes. However, the authors need to state that any conclusion on precursor-product relationships based on bioinformatic analysis remains a hypothesis until verified by actual in vivo lineage-tracing experiments shown later in the manuscript. The term 'in silico lineage tracing' or alike may work. This referee also suggests consolidation of this part of the manuscript. It would be great if the data can be provided to the readers by a publicly accessible database to provide the immense information to the field?
- 2) Experiments testing the scavenging function are interesting (Figure 4c-g) and – to the opinion of this referee – crucial for the definition of a mature macrophage. However, it remains unclear whether dextran particles are engulfed or bound (the term used by the authors) – one is an active process, the other one may be a passive mechanism explaining the observed co-localization. Also, the authors may want to consider using the functional test as baseline for the definition of mature macrophages and, correspondingly, 'label' all other subsets 'non-phagocytic macrophages' (precursors?). The histological analysis needs to be quantified to allow any conclusion to be drawn. Finally, the authors state that MHCII+ cells do not 'bind' dextran, but the picture shows adjacent signals for dextran and MHCII (Figure 4g) – please specify and explain.
- 3) The in vivo lineage-tracing experiments are very elegant and informative. Even though the embryonic origin of CD206+ MHCII- macrophages is shown, the extent of the contribution of embryo-derived macrophages in this population is most likely underestimated. Tissue-specific lineage-tracing cre-transgenic mouse lines are available that exclusively and uniformly label embryo-derived macrophages. The authors could use one of these models or estimate the extent of contribution of embryonic cells to CD206+ MHCII- macrophages by applying TAM pulses at earlier time points (embryonic macrophage progenitors emerge around E8.5). How do the authors exclude the labeling of postnatal CXCR1+ cells through prolonged bioavailability of TAM?
- 4) To show lack of contribution from postnatal definitive HSC-derived cells to testes macrophages the authors should use available lineage-tracing mouse models to substantiate their findings.
- 5) Functional importance of embryonic macrophages for spermatogenesis. The authors deplete macrophages with anti-Csf1r Abs in newborn mice (efficient but transient for CD206+ MHCII- Mp, Figure 8 b-d,f) or use Ccr2-/- mice (inefficient, Figure 6d) and find no effect on spermatogenesis. The time point of macrophage depletion may be too late (NB), and or the depletion not complete preventing a potential role for embryonic macrophages on spermatogenesis to become evident by these assays. A genetic model targeting embryonic macrophages before birth or any other definitive experimental approach would be much more convincing.
- 6) Overall, the manuscript is lengthy and would benefit greatly from consolidation into fewer figures focusing on the crucial experiments.

Minors:

Figure 1a. Insert information on the pre-gating of the cells (same throughout Figure 2).

Figure 4 a vs b. How do the authors explain the discrepancy of frequencies of CD206- MHCII- Mps in the testes of 5 week-old mice between mass and flow cytometry?

Reviewer #2 (Immune metabolism, tissue-resident macrophage)(Remarks to the Author):

Here the authors use a combination of high dimension flow cytometry, fate mapping and depletion studies to examine in detail the origin and phenotypes of macrophages in the testis. They identify CD206 and MHC-II as two key markers for the discrimination of macrophage populations. The analysis is extensive and well presented. The conclusions, however, are largely descriptive. It is of note that they are challenging current dogma and they propose an interesting hypothesis that CD206-MHC-II- cells give rise sequentially to CD206+MHC II- then CD206-MHC-II+ populations. They fairly note that, "These data are compatible with" such a scenario but this is based largely on trajectory analysis of the flow data. Although these analyses have hypothesis generating power, in my opinion this must be further confirmed with transfers, tight fate mapping or other means. Lastly, the fact that depletion of these populations seems to have little if any effect is disappointing. Together with the fact that the only functional assessment is uptake of dextran, Ig complexes or LDL, leaves one thinking that this report is better suited to a highly specialized venue where it will be seen by aficionados of hematopoiesis.

Minor comments:

In figure 1a the distinction seems to be on cd45 density rather than F4/80. In fact many of the assays show differences in CD45 density but this is not commented on in the text.

In contrast to the statements it appears that a substantial number of the F4/80int cells are CD11c+ in the NB and there is a substantial population of eosinophils as well.

Statements such as, "These data are compatible with a scenario that CD206–MHC II– cells infiltrate the testes during the fetal period (presumably as Ly6C+ monocytes) and first give rise to CD206+MHC II– cells in young mice and subsequently differentiate to CD206–MHC II+ macrophages during later postnatal life." Are not sufficiently supported.

The first 3 figures are purely descriptive and could be substantially condensed as in the later figures the populations are largely cooked down to those noted above.

Figure 4 f clearly shows CD206+ cells with no dextran (more than are positive for dextran) yet the analysis seems to show nearly uniform labeling.

Figure 5 is descriptive and needs some sort of quantification if possible.

Reviewer #3 (Remarks to the Author):

Thank you for the opportunity to review the manuscript by Lokka et al.: "Tissue-resident macrophages in the adult testis are of embryonic origin". The authors used mass cytometry, depletion experiments, macrophage deficient mice and fluorescent imaging to investigate the developmental origin and turnover of tissue-resident macrophages in the testis of mice. They define new macrophage cell types,

and show that testicular macrophages are fetal-derived, and are indispensable for spermatogenesis. This is an interesting article that brings new insight into the macrophage biology of the testis and argues against some previously established concepts. The use of mass cytometry and multiple experimental mouse models is a powerful approach. The manuscript could benefit from addressing the following comments.

1. The mass cytometry approach of newborn and postnatal testis identified interesting new cell populations. From a technical standpoint, how did the authors account for sample to sample variability in staining. Were samples from various experimental groups barcoded together? Moreover, it appears that sample groups are being analyzed with different strategies. For example, what is the rationale behind analyzing the newborn data (fig 1f) separately from the postnatal data analysis (fig 2-3). Similarly, why FlowSOM ran only on the postnatal samples and not on the newborn samples?

2. Fetal testis where only assessed in figure 1. All other figures are focused on newborn and/or postnatal testis. It doesn't seem like the assessment of the fetal testis is crucial to the results and conclusions of this manuscript. If it is, this should be made clearer. Moreover, the flow cytometric approach and additional phenotypic assessment of CD115+, CD206+ and Ly6Clow expression seems rather redundant when they had the opportunity to run high-dimensional mass cytometry. Why was decided to not run mass cytometry on embryonic samples?

3. To interrogate the yolk-sac origin of F4/80hi macrophages in fetal testis the authors used a depletion model that should selectively deplete yolk-sac derived macrophage. However, when looking at Suppl. Fig 2e., both F4/80hi and F4/80int cells are present in the brain of the isotype-treated group, with the F4/80int cells remaining after depletion. Why are F4/80int macrophages present in the brain when the authors say there should be solely yolk-sac derived macrophages?

4. In the fate-mapping experiments on line 287-298, YPF-positivity was exclusively found in the CD206+MHCII- cells. However, results show only 1% of the CD206+MHCII- cells are YPF-positive. Consequently, if only 1% of the CD206+MHCII- cells are of yolk-sac origin, it doesn't seem like a surprise that depletion of yolk-sac derived macrophages by anti-CSF1R antibody administration will have an impact on the macrophage populations. This should be discussed in the manuscript.

5. The origin tracking results (Fig 6) show that yolk-sac and fetal-liver derived macrophages contribute to only a part of the tissue-resident macrophages observed in the testis (i.e. there are still macrophages populations present in both experimental models), while bone-marrow derived macrophages do not significantly contribute to the population under normal homeostatic conditions. The authors should speculate where the other tissue-resident macrophages observed in the testis might originate from.

6. An explanation for why the tissue-resident depletion should also be done in newborns next to 10-day-old mice is warranted. In addition, the authors should discuss the difference in obtained results between these 2 experimental models, i.e. recovery of macrophage populations in newborn model while no recovery in 10-day-old mice model.

Reviewers' comments:

Reviewer #1 (Hematopoiesis, tissue-resident macrophage)(Remarks to the Author):

We thank the reviewer for the positive and constructive comments.

Lokka et al. present data on the heterogeneity and origin of testes-resident macrophages and their role in spermatogenesis. Using high dimensional single cell mass cytometry, alterations in macrophage populations from the embryo to adult mice are determined, thereby defining new macrophage subsets. Lineage-tracing and macrophage-depletion studies substantiate the interpretation of the mass cytometry data and the manuscript elegantly bridges descriptive high-dimension mass-cytometry with functional in vivo approaches. The key messages are new and unprecedented to the field, making this a potentially high impact publication. All experiments are well thought through, presented and described. However, a few points need to be addressed to substantiate the strong and important conclusions drawn from the data provided:

1) The mass cytometry data is very fascinating and a lot can be learned about macrophage heterogeneity in the testes. However, the authors need to state that any conclusion on precursor-product relationships based on bioinformatic analysis remains a hypothesis until verified by actual in vivo lineage-tracing experiments shown later in the manuscript. The term 'in silico lineage tracing' or alike may work.

We have further emphasized the hypothesis-generating nature of the bioinformatic analyses by using the nice "in silico lineage tracing" term, as suggested. (lines 128, 184, 196,476,501)

This referee also suggests consolidation of this part of the manuscript.

We have shortened the beginning of the Results section, as requested. The first original 3 figures have now been re-organized into two, and the corresponding results part has been shortened by 30%. (new organization of Figs 1 and 2, and radical shortening of the first two sections of the Results)

It would be great if the data can be provided to the readers by a publicly accessible database to provide the immense information to the field?

The mass cytometry raw data will be deposited to a publicly accessible database (FlowRepository: <https://flowrepository.org>) upon acceptance of the manuscript.

2) Experiments testing the scavenging function are interesting (Figure 4c-g) and – to the opinion of this referee – crucial for the definition of a mature macrophage. However, it remains unclear whether dextran particles are engulfed or bound (the term used by the authors) – one is an active process, the other one may be a passive mechanism explaining the observed co-localization. Also, the authors may want to consider using the functional test as baseline for the definition of mature macrophages and, correspondingly, 'label' all other subsets 'non-phagocytic macrophages' (precursors?).

We used the in vivo scavenging test mainly as a read-out for the accessibility of different macrophage subpopulations to blood-borne ligands (i.e. to define vessel-associated macrophages). This has now been clarified in the text (lines 241-242).

To study the engulfing/endocytic capacity of different macrophages in a more classical way, we performed new ex vivo experiments. We enriched the testis macrophages by magnetic MACS-sorting, and subjected them to fluorescently-labelled high molecular weight dextran. The cells were allowed to engulf the ligand for 1 h at +37 °C (or at 4 °C to control for any passive binding). The cells were then stained, and the endocytosis in each macrophage type was determined using FACS. The endocytosis capacity was then defined by comparing the engulfment at +37 °C to the binding at +4 °C. The results showed that the CD206+ macrophages have the highest endocytic capacity also ex vivo. However, the two other macrophage types were also able to actively engulf dextran thus defining all three populations as bona fide macrophages. (lines 214-217, 670-680, new Fig. 3b)

Please note that we tested in preliminary assays that isolated macrophages from peritoneal cavity are able to engulf dextran without a pre-culturing step, and to avoid possible de-differentiation of the testis macrophages, we subjected them to the dextran right after the sorting. We also reasoned that using a pool of macrophages at the engulfment step would allow direct comparison of the different macrophage subpopulations under the exactly same experimental conditions.

The histological analysis needs to be quantified to allow any conclusion to be drawn.

The quantification of the scavenging data is done using FACS analyses. The histological images here were meant to solely serve as visual evidence that testis macrophages have indeed scavenged the ligands under in situ conditions. We have now explained the rationale (only FACS allows the quantification of high numbers of macrophages, the analyses of the whole of testis tissue volume, and the identification of all three main macrophage populations) more carefully. (lines 220-223)

Finally, the authors state that MHCII+ cells do not 'bind' dextran, but the picture shows adjacent signals for dextran and MHCII (Figure 4g) – please specify and explain.

The dextran signal near the MHCII+ cell does not actually co-localize with the macrophage but to a near-by MHCII-negative cell. This can be seen more clearly in a zoom-in image (the green arrow in the figure below). Please also note that while MHC+ cells are clearly inferior to CD206+ in engulfment of dextran, they still can take-up lower quantities of the cargo (now Fig 3g).

3) The in vivo lineage-tracing experiments are very elegant and informative. Even though the embryonic origin of CD206+ MHCII- macrophages is shown, the extent of the contribution of embryo-derived macrophages in this population is most likely underestimated.

We fully agree that the frequency of the labelled cells is the minimum estimate, since reporter-positive cells never become converted with 100% efficacy in any inducible model. This has now been commented in the text (**lines 258-261**).

Tissue-specific lineage-tracing cre-transgenic mouse lines are available that exclusively and uniformly label embryo-derived macrophages.

To our knowledge there are several good models for exclusive labelling of yolk sac-derived macrophages (such as the CSF1R-reporter, which we have now used in a new set of experiments). Moreover, the literature convincingly shows that E13.5 labelling using the CX3CR1 reporter only labels yolk sac derived macrophages, but not fetal liver derived cells^{1,2}. (**lines 246-258, 521, 539-543, new Figs 4a,b,c and e**)

Unfortunately, we are not aware of any model, which would allow unambiguous selective labelling of all embryonic macrophages (please, see the Table below). In particular, formal lineage tracing of fetal liver-derived macrophages, which constitute a major macrophage type in the postnatal testis, has remained challenging. In addition, timing of conversion in commonly used models like Tie2-Cre and Cx3cr1-Cre is challenging, since early induction will not capture the cells emerging only later during the fetal development, and late induction will not capture the differentiated macrophages, which have already lost the expression of the reporter gene.

Please, note that even the recently published MS3a4 reporter³ for monocyte-derived macrophages does not work in testis (unlike in many other tissues, see Suppl Fig 5B from the paper below), since the authors have shown that for some reason in testis the labelling is almost exclusively seen in non-CD45+ cells (and personal communication with the first author)

[Redacted]

Therefore, we find our analyses of new-born testis as a solid way of defining embryonic-derived cell types, since it is not dependent on the pitfalls of different reporter models.

Model	Labeling results	Ref.
Runx1^{CreER}	Identifies only yolk-sac- derived macrophages. Has very short labeling window during early development.	Hoeffel et al. (2012) ⁴ Ginhoux et al. (2010) ⁵
Csf1r^{CreER}	Useful for identification of yolk-sac- derived macrophages only	Epelman et al. (2014) ⁶ Schultz et al. (2012) ⁷
Flt3^{Cre}	Defines cells that pass through definitive hematopoiesis (HSCs). Does not label some	Boyer et al. (2011) ⁸ Schulz et al. (2012) ⁷

	HSC-derived macrophages during times of rapid expansion (i.e., in early development).	Hoeffel et al. (2015) ⁹ Epelman et al. (2014) ⁶
Cx3cr1^{Cre}	Labels only cells that pass through a CX3CR1 ⁺ stage. Not useful to differentiating between recruited and resident macrophages.	Yona et al. (2013) ¹
Cx3cr1^{CreER}	Limited to tissue macrophages that express CX3CR1 at the time of induction, only transiently labels monocytes.	Yona et al. (2013) ¹ Goldmann et al. (2013) ¹⁰
Tie2^{Mer-iCre-Mer}	An early tamoxifen injection (such as at E7.5) will result in the tagging of all hematopoietic cells emerging before the time of analysis and a late injection (such as at E10.5) will restrict the tagging to only the latest hematopoietic stem cells wave	Busch et al. (2015) ¹¹ Gomez Perdiguero et al. (2015) ¹²
Kit^{MER-iCre-MER}	C-kit is expressed by all hematopoietic progenitors. An early tamoxifen injection (such as at E7.5) will restrict the labeling to early progenitors. However, the fetal liver recruits progenitors of each hematopoietic wave from E8.5 until E11 and these progenitors still express c-kit and coexist after seeding the fetal liver. Model is not suitable to resolve the complexity of the different embryonic hematopoietic waves.	Sheng et al. (2015) ¹³
S100a4^{Cre}	Labels primitive hematopoietic cells and other progenitors early in development. Labels also adult HSCs. Model is not suitable to resolve the complexity of the different embryonic hematopoietic waves.	Hashimoto et al (2013) ¹⁴
LysM^{Cre}	Labels long-term hematopoietic stem cells as well as megakaryocyte/erythrocyte progenitor cells. In addition to monocytes and mature macrophages, LysM is also expressed in most granulocytes and some CD11c ⁺ dendritic cells.	Ye et al (2003) ¹⁵
CD11b^{Cre}	Universal myeloid expression. Model is not suitable to resolve the complexity of the different embryonic hematopoietic waves.	Ferron et al (2005) ¹⁶
F4/80^{Cre}	Expressed to some extent in all macrophages. Model is not suitable to resolve the complexity of the different embryonic hematopoietic waves.	Schaller et al (2002) ¹⁷

The authors could use one of these models or estimate the extent of contribution of embryonic cells to CD206⁺ MHCII⁻ macrophages by applying TAM pulses at earlier time points (embryonic macrophage progenitors emerge around E8.5).

We have now performed new experiments with earlier TAM pulses as requested. When we administered tamoxifen to CSFR1-reporter mice at E8.5, we confirmed the presence of yolk sac-derived cells in the testis postnatally. We also now provide new data with extended follow-up time of yolk sac-derived cells. Notably, we still observed reporter positive cells at week 12, indicating that fetal derived macrophages can persist in testis for a long time. (lines 246-258, 431, 521, 539-543, new Figs 4a,b and e).

How do the authors exclude the labeling of postnatal CXCR1+ cells through prolonged bioavailability of TAM?

It has been extensively shown in the literature that E13.5 labelling in CX3CR1-reporter does not label postnatal cells¹. The tamoxifen concentration is generally thought to remain at a relevant level in mice for 2-3 days after the pulse¹⁸. Thus, it is highly unlikely that any significant conversion would take place in the cells at birth (7 days after the pulse) or later.

4) To show lack of contribution from postnatal definitive HSC-derived cells to testes macrophages the authors should use available lineage-tracing mouse models to substantiate their findings.

Unfortunately, we are not aware of any lineage-tracing model, which would allow selective labelling of postnatal HSC-derived cells. Please, see the Table above. In our re-population studies, we find no role for CCR2 in the re-population, we do not find any increase of Ly6C+ monocytes in the testis, and we do not find any proliferating monocytes in the testis. Collectively these strongly argue against a contribution of postnatal definitive HSC to testis macrophages.

5) Functional importance of embryonic macrophages for spermatogenesis. The authors deplete macrophages with anti-Csf1r Abs in newborn mice (efficient but transient for CD206+ MHCII- Mp, Figure 8 b-d,f) or use Ccr2-/- mice (inefficient, Figure 6d) and find no effect on spermatogenesis. The time point of macrophage depletion may be too late (NB), and or the depletion not complete preventing a potential role for embryonic macrophages on spermatogenesis to become evident by these assays. A genetic model targeting embryonic macrophages before birth or any other definitive experimental approach would be much more convincing.

We have now experimentally addressed this important point using several new approaches.

- 1) We performed new assays combining genetic and pharmacological depletion of macrophages postnatally. In these experiments, CCR2-/- mice were injected with CSF1 antibody at birth. Again we saw a major reduction of testis macrophages, but apparently normal spermatogenesis. (lines 356-376, 563, new Fig. 6 a,b,c,e,f, new Supplementary Figs 9h,i, 11e,f)
- 2) To perform more complete and earlier macrophage depletion, we developed a new pharmacological depletion protocol. We treated E6.5 mice using CD115 antibody (to deplete yolk sac-derived cells), then treated the new-born mice with CSF1 antibody (to deplete all macrophages) and finally re-treated the 2 wk old mice with CSF1-antibody (to again deplete any possibly remaining macrophages). This depletion regimen caused practically complete lack of testis macrophages at week 5. Notably, in this model we saw a clear effect on testicular organization and spermatogenesis (lines 384-408, 565-573, new Figs 7b,c,d,e,f,g,h,i; new Supplementary Figures 10b, 11i,j).
- 3) Finally, we used CSF1^{op} mice, which lack a critical macrophage growth factor CSF1 throughout the development. We found a complete lack of all testicular macrophage subtypes in these mice, and a concomitant substantial impairment of spermatogenesis. (lines 378-383, 391-408, 521, new Figs 7a, d,e,f,g,h,i, new Supplemental Figs 10b, 11g,h)

These important new experiments led us to modify our conclusions: Apparently the presence of macrophages postnatally is not critical to spermatogenesis, while their presence during the fetal development is instrumental. (lines 46-47, 51, 405-408, 421-422, 491-495)

6) Overall, the manuscript is lengthy and would benefit greatly from consolidation into fewer figures focusing on the crucial experiments.

We have shortened the manuscript and the number of main figures radically (from 9 to 7) even though extensive new experimentation is now included.

Minors:

Figure 1a. Insert information on the pre-gating of the cells (same throughout Figure 2).

The pre-gating for all analyses is shown in Supplemental Figure 1. This is now indicated more clearly throughout the text (lines 94,119,144,211,311,328; amended Supplemental Fig 1a-h).

Figure 4 a vs b. How do the authors explain the discrepancy of frequencies of CD206- MHCII- Mps in the testes of 5 week-old mice between mass and flow cytometry?

Re-adjustment of the gating in the mass cytometry experiments to mimic more closely the one used in FACS diminished (but did not abolish) the difference in the frequencies of the CD206-MHC- cells detected by these two techniques. We think that the remaining difference is likely caused by the different methods for excluding non-viable cells and duplets, and in the performance of different primary antibody clones (e.g. F4/80 clone CI:A3-1 in FACS and BM8 in mass cytometry).

General notion: It has been quite challenging to perform new experiments under the current COVID-19 situation. Our university has been closed, and we have needed to justify the “critical functions” nature of these experiments to allow the entrance to the lab. Moreover, it would have been impossible to obtain any potential new reporter mouse lines from abroad to be used within the allocated 3 month revision time.

1. Yona, S. *et al.* Fate mapping reveals origins and dynamics of monocytes and tissue macrophages under homeostasis. *Immunity* **38**, 79-91 (2013).
2. Molawi, K. *et al.* Progressive replacement of embryo-derived cardiac macrophages with age. *J Exp Med* **211**, 2151-2158 (2014).
3. Liu, Z. *et al.* Fate Mapping via Ms4a3-Expression History Traces Monocyte-Derived Cells. *Cell* **178**, 1509-1525.e1519 (2019).
4. Hoeffel, G. *et al.* Adult Langerhans cells derive predominantly from embryonic fetal liver monocytes with a minor contribution of yolk sac-derived macrophages. *J Exp Med* **209**, 1167-1181 (2012).
5. Ginhoux, F. *et al.* Fate mapping analysis reveals that adult microglia derive from primitive macrophages. *Science* **330**, 841-845 (2010).
6. Epelman, S. *et al.* Embryonic and adult-derived resident cardiac macrophages are maintained through distinct mechanisms at steady state and during inflammation. *Immunity* **40**, 91-104 (2014).

7. Schulz, C. *et al.* A lineage of myeloid cells independent of Myb and hematopoietic stem cells. *Science* **336**, 86-90 (2012).
8. Boyer, S.W., Beaudin, A.E. & Forsberg, E.C. Mapping differentiation pathways from hematopoietic stem cells using Flk2/Flt3 lineage tracing. *Cell Cycle* **11**, 3180-3188 (2012).
9. Hoeffel, G. *et al.* C-Myb(+) erythro-myeloid progenitor-derived fetal monocytes give rise to adult tissue-resident macrophages. *Immunity* **42**, 665-678 (2015).
10. Goldmann, T. *et al.* A new type of microglia gene targeting shows TAK1 to be pivotal in CNS autoimmune inflammation. *Nat Neurosci* **16**, 1618-1626 (2013).
11. Busch, K. *et al.* Fundamental properties of unperturbed haematopoiesis from stem cells in vivo. *Nature* **518**, 542-546 (2015).
12. Gomez Perdiguero, E. *et al.* Tissue-resident macrophages originate from yolk-sac-derived erythro-myeloid progenitors. *Nature* **518**, 547-551 (2015).
13. Sheng, J., Ruedl, C. & Karjalainen, K. Most Tissue-Resident Macrophages Except Microglia Are Derived from Fetal Hematopoietic Stem Cells. *Immunity* **43**, 382-393 (2015).
14. Hashimoto, D. *et al.* Tissue-resident macrophages self-maintain locally throughout adult life with minimal contribution from circulating monocytes. *Immunity* **38**, 792-804 (2013).
15. Ye, M. *et al.* Hematopoietic stem cells expressing the myeloid lysozyme gene retain long-term, multilineage repopulation potential. *Immunity* **19**, 689-699 (2003).
16. Ferron, M. & Vacher, J. Targeted expression of Cre recombinase in macrophages and osteoclasts in transgenic mice. *Genesis* **41**, 138-145 (2005).
17. Schaller, E. *et al.* Inactivation of the F4/80 glycoprotein in the mouse germ line. *Mol Cell Biol* **22**, 8035-8043 (2002).
18. Jahn, H.M. *et al.* Refined protocols of tamoxifen injection for inducible DNA recombination in mouse astroglia. *Sci Rep* **8**, 5913 (2018).

Reviewer #2 (Immune metabolism, tissue-resident macrophage)(Remarks to the Author):

We thank the reviewer for the insightful comments.

Here the authors use a combination of high dimension flow cytometry, fate mapping and depletion studies to examine in detail the origin and phenotypes of macrophages in the testis. They identify CD206 and MHC-II as two key markers for the discrimination of macrophage populations. The analysis is extensive and well presented. The conclusions, however, are largely descriptive. It is of note that they are challenging current dogma and they propose an interesting hypothesis that CD206-MHC-II- cells give rise sequentially to CD206+MHC II- then CD206-MHC-II+ populations. They fairly note that, "These data are compatible with" such a scenario but this is based largely on trajectory analysis of the flow data. Although these analyses have hypothesis generating power, in my opinion this must be further confirmed with transfers, tight fate mapping or other means.

We have further emphasized the hypothesis-generating nature of the bioinformatic analyses by using the "in silico lineage tracing" term. (lines 128, 184, 196,476,501)

We have now performed new kinetic experiments using the fate-mapping models. Specifically, we have:

-Used *Csf1*^{Mer-iCre-Mer} as a new fate-mapping tool. When we administered tamoxifen to CSF1R mice at E8.5, we confirmed the presence of yolk sac-derived cells in the F4/80High macrophage population at E16.5 and in all three major testicular macrophage populations 5 wk postnatally. (lines 246-253, 431, 521, 539-541, new Figs 4a,b)

- We now provide new data with extended follow-up time of yolk sac-derived cells. Notably, we still observed reporter positive cells at week 10, indicating that fetal derived macrophages can persist in testis for a long time. (lines 257-258, new Fig. 4e)

-We have performed in vivo BrdU-labelings to follow the cell proliferation in different populations during the re-population experiments. (lines 336-341, 615-618, new Fig 5i, new Supplemental Fig. 9g).

-We have refined Ki67-based cell proliferation analyses by in silico trajectory analyses, which show clear and quite linear connectivity pathway between fetal monocytes, CD206-MHCII- cells and CD206+MHCII- cells in young mice. (lines 192-196; new Supplementary Fig 3c).

-we have more rigorously excluded the contribution of CCR2-mediated migration of bone marrow – derived monocytes into experimentally emptied testicular mice by showing that the delayed and partial recovery of testicular macrophages is similar in wild-type mice and CCR2-/- mice depleted of macrophages by a single anti-CSF1 antibody injection at birth. (new Supplementary Figs 9h,i).

All our new experimental data are compatible with the possibility that CD206+ testicular macrophages are generated both as direct descendants of yolk sac macrophages and by fetal liver derived monocytes. In the latter case these monocytes first differentiate into CD206-MHC-macrophages, and subsequently to interstitial CD206+MHCII- macrophages and CD206-MHCII+ peritubular macrophages (without an input from bone marrow derived monocytes).

We find it as a strength of our article that our data challenge the widely advocated view on testicular macrophage origin, which was formulated largely based on descriptive FACS stainings with a limited number of markers. Notably, we have tested and observed that the cell transfer model used by Mossadegh-Keller et al.¹ is by no means specific to bone marrow derived cells, since the labelled HSC

also produce monocytes in the newborn liver. This is a good illustration of the technical difficulties the whole field is facing. Moreover, to our knowledge there are no tight fate mapping models for labelling fetal liver derived macrophages.

Model	Labeling results	Ref.
Runx1^{CreER}	Identifies only yolk-sac- derived macrophages. Has very short labeling window during early development.	Hoeffel et al. (2012) ² Ginhoux et al. (2010) ³
Csf1r^{CreER}	Useful for identification of yolk-sac- derived macrophages only	Epelman et al. (2014) ⁴ Schultz et al. (2012) ⁵
Flt3^{Cre}	Defines cells that pass through definitive hematopoiesis (HSCs). Does not label some HSC-derived macrophages during times of rapid expansion (i.e., in early development).	Boyer et al. (2011) ⁶ Schulz et al. (2012) ⁵ Hoeffel et al. (2015) ⁷ Epelman et al. (2014) ⁴
Cx3cr1^{Cre}	Labels only cells that pass through a CX3CR1 ⁺ stage. Not useful to differentiating between recruited and resident macrophages.	Yona et al. (2013) ⁸
Cx3cr1^{CreER}	Limited to tissue macrophages that express CX3CR1 at the time of induction, only transiently labels monocytes.	Yona et al. (2013) ⁸ Goldmann et al. (2013) ⁹
Tie2^{Mer-iCre-Mer}	An early tamoxifen injection (such as at E7.5) will result in the tagging of all hematopoietic cells emerging before the time of analysis and a late injection (such as at E10.5) will restrict the tagging to only the latest hematopoietic stem cells wave	Busch et al. (2015) ¹⁰ Gomez Perdiguero et al. (2015) ¹¹
Kit^{MER-iCre-MER}	C-kit is expressed by all hematopoietic progenitors. An early tamoxifen injection (such as at E7.5) will restrict the labeling to early progenitors. However, the fetal liver recruits progenitors of each hematopoietic wave from E8.5 until E11 and these progenitors still express c-kit and coexist after seeding the fetal liver. Model is not suitable to resolve the complexity of the different embryonic hematopoietic waves.	Sheng et al. (2015) ¹²
S100a4^{Cre}	Labels primitive hematopoietic cells and other progenitors early in development. Labels also adult HSCs. Model is not suitable to resolve the complexity of the different embryonic hematopoietic waves.	Hashimoto et al (2013) ¹³
LysM^{Cre}	Labels long-term hematopoietic stem cells as well as megakaryocyte/erythrocyte progenitor cells. In addition to monocytes and mature macrophages, LysM is also expressed in most granulocytes and some CD11c ⁺ dendritic cells.	Ye et al (2003) ¹⁴

CD11b^{Cre} *Universal myeloid expression.* Model is not suitable to resolve the complexity of the different embryonic hematopoietic waves. Ferron et al (2005)¹⁵

F4/80^{Cre} *Expressed to some extent in all macrophages.* Model is not suitable to resolve the complexity of the different embryonic hematopoietic waves. Schaller et al (2002)¹⁶

Please, note that even the recently published MS3a4 reporter¹⁷ for monocyte-derived macrophages does not work in testis (unlike in many other tissues, see Suppl Fig 5B from the paper below), since the authors have shown that for some reason in testis the labelling is almost exclusively seen in non-CD45+ cells (and personal communication with the first author)

[Redacted]

Therefore, we find our analyses of new-born testis as a solid way of defining

Lastly, the fact that depletion of these populations seems to have little if any effect is disappointing.

We have now performed new functional experiments, which showed that prenatal macrophages (unlike macrophages postnatally) play a functional role in regulating male fertility .

- 1) We performed new assays combining genetic and pharmacological depletion of macrophages postnatally. In these experiments, CCR2^{-/-} mice were injected with CSF1 antibody at birth. Again we saw a major reduction of testis macrophages, but apparently normal spermatogenesis. **(lines 356-376, 563, new Fig. 6 a,b,c,e,f, new Supplementary Figs 9h,i, 11e,f)**
- 2) To perform more complete and earlier macrophage depletion, we developed a new pharmacological depletion protocol. We treated E6.5 mice using CD115 antibody (to deplete yolk sac-derived cells), then treated the new-born mice with CSF1 antibody (to deplete all macrophages) and finally re-treated the 2 wk old mice with CSF-1antibody (to again deplete any possibly remaining macrophages). This depletion regimen caused practically complete lack of testis macrophages at week 5. Notably, in this model we saw a clear effect on testicular organization and spermatogenesis **(lines 384-408, 565-573, new Figs 7b,c,d,e,f,g,h,i; new Supplementary Figures 10b, 11i,j).**
- 3) Finally, we used CSF1^{op} mice, which lack a critical macrophage growth factor CSF1 throughout the development. We found a complete lack of all testicular macrophage subtypes in these mice, and a concomitant substantial impairment of spermatogenesis.**(lines 378-383, 391-408, 521, new Figs 7a, d,e,f,g,h,i, new Supplemental Figs 10b, 11g,h)**

These important new experiments led us to modify our conclusions: Apparently the presence of macrophages postnatally is not critical to spermatogenesis, while their presence during the fetal development is instrumental. (lines 46-47, 51, 405-408, 421-422, 491-495)

Together with the fact that the only functional assessment is uptake of dextran, Ig complexes or LDL, leaves one thinking that this report is better suited to a highly specialized venue where it will be seen by aficionados of hematopoiesis.

Minor comments:

In figure 1a the distinction seems to be on cd45 density rather than F4/80. In fact many of the assays show differences in CD45 density but this is not commented on in the text.

The reviewer is absolutely right. The same observation has been done by others e.g. in brain macrophages⁷ We have now pointed this out in the text (lines 99-100) .

In contrast to the statements it appears that a substantial number of the F4/80int cells are CD11c+ in the NB and there is a substantial population of eosinophils as well.

We apologize for the imprecise wording in the original version. Dendritic cells are typically CD11High and eosinophils SiglecFHigh, while low levels of these antigens can be expressed on other cell types also. We have now re-worded the text (lines 105-107).

Statements such as, "These data are compatible with a scenario that CD206–MHC II– cells infiltrate the testes during the fetal period (presumably as Ly6C+ monocytes) and first give rise to CD206+MHC II– cells in young mice and subsequently differentiate to CD206–MHC II+ macrophages during later postnatal life." Are not sufficiently supported.

We have now further emphasized the experimental basis of these conclusions and clearly state that they need to be confirmed when suitable models become available. Please, also see above for the additional experimental models that we have done for the revised manuscript. (lines 507-509)

The first 3 figures are purely descriptive and could be substantially condensed as in the later figures the populations are largely cooked down to those noted above.

We have now condensed the 3 first figures into two smaller ones, as requested. We have also radically shortened the corresponding parts of the text.

Figure 4 f clearly shows CD206+ cells with no dextran (more than are positive for dextran) yet the analysis seems to show nearly uniform labeling.

The quantitation of the scavenging data was done using FACS. The histological images were meant to solely serve as visual evidence that testis macrophages have indeed scavenged the ligands under in situ conditions. We have now explained the rationale (only FACS allows the quantification of high numbers of macrophages, the analyses of the whole of testis tissue volume, and the identification of all three main macrophage populations) in more detail. Please note that each dot in the FACS quantification represents an individual mouse. MFI for each mouse has been calculated from hundreds of cells, and among individual cells the intensity of dextran signal varies considerably. In

addition, we have now replaced the representative histological image with a new one which better illustrates the average of the quantitative data. (lines 220-223, new Fig. 3f)

Figure 5 is descriptive and needs some sort of quantification if possible.

We have now quantified the distances of macrophages to the nearest vessels, as requested. These data show that over 80% of the vessel-associated macrophages are CD206+ cells. (lines 233-235, 753-757)

General notion: It has been quite challenging to perform new experiments under the current COVID-19 situation. Our university has been closed, and we have needed to justify the “critical functions” nature of these experiments to allow the entrance to the lab. Moreover, it would have been impossible to obtain any potential new reporter mouse lines from abroad to be used within the allocated 3 month revision time.

1. Mossadegh-Keller, N. *et al.* Developmental origin and maintenance of distinct testicular macrophage populations. *J Exp Med* **214**, 2829-2841 (2017).
2. Hoeffel, G. *et al.* Adult Langerhans cells derive predominantly from embryonic fetal liver monocytes with a minor contribution of yolk sac-derived macrophages. *J Exp Med* **209**, 1167-1181 (2012).
3. Ginhoux, F. *et al.* Fate mapping analysis reveals that adult microglia derive from primitive macrophages. *Science* **330**, 841-845 (2010).
4. Epelman, S. *et al.* Embryonic and adult-derived resident cardiac macrophages are maintained through distinct mechanisms at steady state and during inflammation. *Immunity* **40**, 91-104 (2014).
5. Schulz, C. *et al.* A lineage of myeloid cells independent of Myb and hematopoietic stem cells. *Science* **336**, 86-90 (2012).
6. Boyer, S.W., Beaudin, A.E. & Forsberg, E.C. Mapping differentiation pathways from hematopoietic stem cells using Flk2/Flt3 lineage tracing. *Cell Cycle* **11**, 3180-3188 (2012).
7. Hoeffel, G. *et al.* C-Myb(+) erythro-myeloid progenitor-derived fetal monocytes give rise to adult tissue-resident macrophages. *Immunity* **42**, 665-678 (2015).
8. Yona, S. *et al.* Fate mapping reveals origins and dynamics of monocytes and tissue macrophages under homeostasis. *Immunity* **38**, 79-91 (2013).
9. Goldmann, T. *et al.* A new type of microglia gene targeting shows TAK1 to be pivotal in CNS autoimmune inflammation. *Nat Neurosci* **16**, 1618-1626 (2013).
10. Busch, K. *et al.* Fundamental properties of unperturbed haematopoiesis from stem cells in vivo. *Nature* **518**, 542-546 (2015).
11. Gomez Perdiguero, E. *et al.* Tissue-resident macrophages originate from yolk-sac-derived erythro-myeloid progenitors. *Nature* **518**, 547-551 (2015).
12. Sheng, J., Ruedl, C. & Karjalainen, K. Most Tissue-Resident Macrophages Except Microglia Are Derived from Fetal Hematopoietic Stem Cells. *Immunity* **43**, 382-393 (2015).
13. Hashimoto, D. *et al.* Tissue-resident macrophages self-maintain locally throughout adult life with minimal contribution from circulating monocytes. *Immunity* **38**, 792-804 (2013).
14. Ye, M. *et al.* Hematopoietic stem cells expressing the myeloid lysozyme gene retain long-term, multilineage repopulation potential. *Immunity* **19**, 689-699 (2003).
15. Ferron, M. & Vacher, J. Targeted expression of Cre recombinase in macrophages and osteoclasts in transgenic mice. *Genesis* **41**, 138-145 (2005).

16. Schaller, E. *et al.* Inactivation of the F4/80 glycoprotein in the mouse germ line. *Mol Cell Biol* **22**, 8035-8043 (2002).
17. Liu, Z. *et al.* Fate Mapping via Ms4a3-Expression History Traces Monocyte-Derived Cells. *Cell* **178**, 1509-1525.e1519 (2019).

Reviewer #3 (Remarks to the Author):

We thank the reviewer for the positive and constructive comments.

Thank you for the opportunity to review the manuscript by Lokka et al.: “Tissue-resident macrophages in the adult testis are of embryonic origin”. The authors used mass cytometry, depletion experiments, macrophage deficient mice and fluorescent imaging to investigate the developmental origin and turnover of tissue-resident macrophages in the testis of mice. They define new macrophage cell types, and show that testicular macrophages are fetal-derived, and are indispensable for spermatogenesis. This is an interesting article that brings new insight into the macrophage biology of the testis and argues against some previously established concepts. The use of mass cytometry and multiple experimental mouse models is a powerful approach. The manuscript could benefit from addressing the following comments.

1. The mass cytometry approach of newborn and postnatal testis identified interesting new cell populations. From a technical standpoint, how did the authors account for sample to sample variability in staining. Were samples from various experimental groups barcoded together?

Our data are from individual mice or from pools of mice (as indicated in the Figure legends and Methods), and we have not used barcoding. Instead, we have performed sufficient number of repetitions so that the means reveal the biological results apart from sample-to-sample variability.

Moreover, it appears that sample groups are being analyzed with different strategies. For example, what is the rationale behind analyzing the newborn data (fig 1f) separately from the postnatal data analysis (fig 2-3).

The cell numbers in embryonic samples did not allow mass cytometric analyses. Even when we pooled 10 testes from 5 mice, barely enough events were found in new-borns. Moreover, since there is only one day difference between the newborns and fetal life, we used the same analyses for them (apart from mass cytometry). The 2-12 wk time points, on the other hand, constitute a postnatal continuum which was possible to follow in much greater detail by mass cytometry. (lines 116-117)

Similarly, why FlowSOM ran only on the postnatal samples and not on the newborn samples?

Please see above for the challenge with the cell numbers. Of note, with the postnatal samples we saw that manual gating and FlowSOM gave practically identical results.

2. Fetal testis where only assessed in figure 1. All other figures are focused on newborn and/or postnatal testis. It doesn't seem like the assessment of the fetal testis is crucial to the results and conclusions of this manuscript. If it is, this should be made clearer. Moreover, the flow cytometric approach and additional phenotypic assessment of CD115+, CD206+ and Ly6Clow expression seems rather redundant when they had the opportunity to run high-dimensional mass cytometry. Why was decided to not run mass cytometry on embryonic samples?

The fetal data are very critical for the manuscript. They show the early infiltration of macrophages to testis, and the dual origin (yolk sac and fetal liver) of those cells, and thereby alter the existing dogma. This has now been made clearer. (lines 133-139)

Please, see above the reasons for not running mass cytometry on fetal samples. That is why we tried to include at least some important markers (such as CD115, CD206 and Ly6C) in the FACS panels as well.

3. To interrogate the yolk-sac origin of F4/80hi macrophages in fetal testis the authors used a depletion model that should selectively deplete yolk-sac derived macrophage. However, when looking at Suppl. Fig 2e., both F4/80hi and F4/80int cells are present in the brain of the isotype-treated group, with the F4/80int cells remaining after depletion. Why are F4/80int macrophages present in the brain when the authors say there should be solely yolk-sac derived macrophages?

The phenomenon pointed out by the reviewer has been noticed before by us and others¹. The likely explanation is that when brain is experimentally emptied from the normal yolk sac-derived macrophages, monocyte-dependent cells (F4/80 int) can later start to infiltrate the area to compensate for the original cells.

4. In the fate-mapping experiments on line 287-298, YPF-positivity was exclusively found in the CD206+MHCII- cells. However, results show only 1% of the CD206+MHCII- cells are YPF-positive. Consequently, if only 1% of the CD206+MHCII- cells are of yolk-sac origin, it doesn't seem like a surprise that depletion of yolk-sac derived macrophages by anti-CSF1R antibody administration will have an impact on the macrophage populations. This should be discussed in the manuscript.

We agree. Importantly, tamoxifen induction never gives 100% conversion efficacy in any reporter model. Therefore, the percentage is certainly an underestimate of the real situation. This has been explained in more detail, as requested. (lines 258-260)

5. The origin tracking results (Fig 6) show that yolk-sac and fetal-liver derived macrophages contribute to only a part of the tissue-resident macrophages observed in the testis (i.e. there are still macrophages populations present in both experimental models), while bone-marrow derived macrophages do not significantly contribute to the population under normal homeostatic conditions. The authors should speculate where the other tissue-resident macrophages observed in the testis might originate from.

The reviewer interprets the results correctly. We think that the depletion efficacy of yolk sac-derived cells is almost complete in our experiments, and we can conclude that only few of these macrophages persist normally in postnatal testis. In contrast, fetal-liver derived macrophages are not absent, but rather strongly reduced, in *Plvap*^{-/-} model. Therefore, it is possible that the remaining fetal liver –derived macrophages expand in the testis and produce the observed populations during the re-population. Moreover, our data show that bone marrow derived cells do not infiltrate normal testis via CCR2-dependent (or Nur77-dependent) mechanisms. Moreover, we do not find proliferating monocytes in testes during re-population in our new BrdU-labelling experiments. Our old and new measurements of cell division thus support the notion of local macrophage proliferation in testis, and the virtual absence of incoming bone marrow derived monocytes. This has now been explained more thoroughly in the text. (lines 336-341, 346-351, new Fig. 5i, new Supplemental Fig.3c (Ki67 trajectories)

6. An explanation for why the tissue-resident depletion should also be done in newborns next to 10-day-old mice is warranted. In addition, the authors should discuss the difference in obtained results between these 2 experimental models, i.e. recovery of macrophage populations in newborn model while no recovery in 10-day-old mice model.

The reason for doing tissue-depletion also in new born was timing. We suspected that macrophages could be important for the testis function only transiently during the first week (or 10d) of postnatal life. Therefore, we wanted to ensure that there are no testis macrophages present postnatally by depleting the cells immediately after birth.

We have now gone further and experimentally addressed this important point using several new approaches.

- 1) We performed new assays combining genetic and pharmacological depletion of macrophages postnatally. In these experiments, CCR2^{-/-} mice were injected with CSF1 antibody at birth. Again we saw a major reduction of testis macrophages, but apparently normal spermatogenesis. **(lines 356-376, 563, new Fig. 6 a,b,c,e,f, new Supplementary Figs 9h,i, 11e,f)**
- 2) To perform more complete and earlier macrophage depletion, we developed a new pharmacological depletion protocol. We treated E6.5 mice using CD115 antibody (to deplete yolk sac-derived cells), then treated the new-born mice with CSF1 antibody (to deplete all macrophages) and finally re-treated the 2 wk old mice with CSF1-antibody (to again deplete any possibly remaining macrophages). This depletion regimen caused practically complete lack of testis macrophages at week 5. Notably, in this model we saw a clear effect on testicular organization and spermatogenesis **(lines 384-408, 565-573, new Figs 7b,c,d,e,f,g,h,i; new Supplementary Figures 10b, 11i,j)**.
- 3) Finally, we used CSF1^{op} mice, which lack a critical macrophage growth factor CSF1 throughout the development. We found a complete lack of all testicular macrophage subtypes in these mice, and a concomitant substantial impairment of spermatogenesis. **(lines 378-383, 391-408, 521, new Figs 7a, d,e,f,g,h,i, new Supplemental Figs 10b, 11g,h)**

These important new experiments led us to modify our conclusions: Apparently the presence of macrophages postnatally is not critical to spermatogenesis, while their presence during the fetal development is instrumental. **(lines 46-47, 51, 405-408, 421-422, 491-495)**

REF.

1. Hoeffel, G. *et al.* C-Myb(+) erythro-myeloid progenitor-derived fetal monocytes give rise to adult tissue-resident macrophages. *Immunity* **42**, 665-678 (2015).

REVIEWERS' COMMENTS:

Reviewer #1 (Remarks to the Author):

While this referee based on own but not published experiments respectfully disagrees on the lack of bioavailability of TAM at later time points of development when administered at E13.5 the new labeling experiments performed at E8.5 answer my questions.

Lineage tracing of EMP-derived tissue-resident macrophages is possible by using *Tnfrsf11a-cre* (*Rank-cre*) knock in mice (Percin et al 2018 Nature Communications, Jacome-Galarza et al 2019 Nature). To label macrophage progeny of definitive HSCs *Vav-cre* (Jacome-Galarza et al 2019 Nature) or *Flk2/Flt3-cre* (Gomez Perdiguero et al 2015 Nature) lineage-tracing mouse lines can be used.

The new approaches for depleting embryonic macrophages significantly substantiate the new conclusions drawn by the authors! In line with this comment the authors have sufficiently addressed and resolved all my concerns.

Reviewer #2 (Remarks to the Author):

I find the revised manuscript to be substantially improved. The additional data regarding function of these cells in development is interesting and well done. The authors did a great job of addressing my concerns.

Dan McVicar
Laboratory of Cancer ImmunoMetabolism, NCI

Reviewer #3 (Remarks to the Author):

I have reviewed the revised manuscript by Salmi et al. The authors have adequately addressed my comments with revisions to the text and additional experimental evidence. Technical limitations (low cell counts in younger animals) were also made more clear. The revised article is much improved and I have no further comments.